# Low-field onset of Wannier-Stark localization in a polycrystalline hybrid organic inorganic perovskite

Daniel Berghoff[1,4], Johannes Bühler[2,4], Mischa Bonn [3], Alfred Leitenstorfer [2], Torsten Meier [1✉] & Heejae Kim [3✉]

Methylammonium lead iodide perovskite (MAPbI$_3$) is renowned for an impressive power conversion efficiency rise and cost-effective fabrication for photovoltaics. In this work, we demonstrate that polycrystalline MAPbI$_3$s undergo drastic changes in optical properties at moderate field strengths with an ultrafast response time, via transient Wannier Stark localization. The distinct band structure of this material - the large lattice periodicity, the narrow electronic energy bandwidths, and the coincidence of these two along the same high-symmetry direction – enables relatively weak fields to bring this material into the Wannier Stark regime. Its polycrystalline nature is not detrimental to the optical switching performance of the material, since the least dispersive direction of the band structure dominates the contribution to the optical response, which favors low-cost fabrication. Together with the outstanding photophysical properties of MAPbI$_3$, this finding highlights the great potential of this material in ultrafast light modulation and novel photonic applications.

[1] Department of Physics, Paderborn University, D-33098 Paderborn, Germany. [2] Department of Physics and Center for Applied Photonics, University of Konstanz, D-78457 Konstanz, Germany. [3] Department of Molecular Spectroscopy, Max Planck Institute for Polymer Research, D-55128 Mainz, Germany. [4]These authors contributed equally: Daniel Berghoff, Johannes Bühler. ✉email: torsten.meier@uni-paderborn.de; kim@mpip-mainz.mpg.de

Methylammonium lead iodide perovskite (MAPbI$_3$) has become a remarkable material for photovoltaic applications due to the dramatic increase of the power conversion efficiency[1] and the cost-effective fabrication processes[2]. The success of this material is due to its large absorption cross-section[3] and the exceptional transport properties such as long carrier diffusion lengths[4,5], high carrier mobilities[6], and defect tolerance[7].

Besides their use in solar cells and light-emitting diodes, in this work, we demonstrate that MAPbI$_3$ also has outstanding properties as a promising optical modulator. Optical modulation is a crucial function for photonic and optoelectronic applications, such as optical data interconnect, optical information processing, environmental monitoring, biosensing, medicine, and security applications[8]. As these technologies are increasingly demanding fast, efficient, and broadband optical modulators, one of the essential properties is a substantial change of the absorption edge with a high modulation rate and a relatively modest energy. We demonstrate that 38% modulation depth and <20 fs response time can be achieved with solution-processed, polycrystalline MAPbI$_3$, via all-optical operation type with weak biasing fields. The key figures of merit include large modulation depth with an ultrafast response time (limited only by the interband dephasing time), large-scale fabrication with low cost, broad wavelength coverage in visible region (1.4 ~ 2.4 eV at least), and almost no limitation in size or fabrication protocol, which is suitable for emerging flexible and/or compact devices. Whereas conventional semiconductors constituting photo-detectors, e.g., Si or InGaAs, require costly manufacturing processes and are limited to traditional rigid-type devices, perovskites with distinct crystal structures exhibit ultrafast response (sub-20 fs), while simultaneously supporting cheap and flexible polycrystalline film fabrication.

The mechanism of this optical modulation, the Wannier–Stark localization, is one of novel states of matter in the presence of strong electric fields[9]. In the presence of strong external electric fields $E$, the continuum of electronic energy bands splits into a series of discrete levels in the direction of the field[9] and the corresponding wavefunctions are confined on a length scale given by $\Delta/(eE)$, where $\Delta$ is the energetic width of the electronic band in the absence of biasing. These localized states, the Wannier–Stark states[10,11], are equally spaced both in energy by an amount $eED$ and in space by the lattice period $D$. As a spatial separation of $nD$ lattice periods results in an energy shift of $neED$ with respect to the central spatially direct ($n = 0$) transition, this Wannier–Stark localization leads to strong spectral modulation of the interband absorption continuum below and above the optical bandgap.

Following the initial observations in semiconductor superlattices under static bias fields[8–10] Wannier–Stark ladders have been proposed and realized in various physical systems featuring wave propagation in the presence of periodic potentials and a homogeneous force. Examples include ultracold atoms in an accelerating one-dimensional (1D) standing wave[12], waveguide arrays with linearly varying propagation constants[13], and self-accelerating optical beams in 1D photonic lattice[14]. Several fundamental observations and device applications from the Wannier–Stark localization have been focused on statically biased artificial semiconductor superlattices[15–19]. However, in natural homogeneous solids, where the periodicity is dictated by the atomic structure, such an extreme state of matter has never been achieved using static biasing. To resolve optical transitions to individual Wannier–Stark states in, e.g., absorption spectra, their energetic spacing needs to be larger than the (total) linewidth $\Gamma$, i.e., $eED > \Gamma$[16,17,20]. Due to the small lattice constant of bulk crystals and the large linewidth, which results from the scattering of electrons with lattice vibrations and other electrons, the requirement $eED > \Gamma$ can typically not be fulfilled under stationary external fields below the strength where the dielectric breakdown occurs[18,19]. So far, only one natural solid, a single crystal GaAs[21], has allowed for achieving the Wannier–Stark localization transiently by virtue of the recent availability of extremely intense and phase-stable pulses of multi-terahertz radiation[22,23]. The ultrafast biasing fields could reach amplitudes up to several tens of MV/cm[22,23], i.e., field strengths comparable to the interatomic fields. For GaAs, an optimally oriented single crystal was required to observe Wannier–Stark localization with the required field strengths exceeding 10 MV/cm[21].

However, we observe here that a disordered, solution-processed, polycrystalline film of MAPbI$_3$ (Fig. 1a) undergoes the transient Wannier–Stark localization at a substantially lower field strength. Already at relatively modest field strengths, the thin film's optical transmission is modified by tens of percent. This material exhibits a tetragonal structure with lattice parameters of $a = 8.8$ Å and $c = 12.5$ Å at room temperature by the expansion of the cubic perovskite unit cell[24,25]. The periodicities are nearly twice as large as the lattice parameter $a = 5.6$ Å of cubic GaAs[21]. We will show that the large relevant lattice constant (Fig. 1a), the small width of electronic energy bands (Fig. 1b), and the coincidence of these two along the same high-symmetry direction lead to Stark localization in this organic perovskite at field amplitudes as low as 3 MV/cm, i.e., at a fraction of the field strength required to enter this regime in optimally oriented, single-crystalline GaAs. Moreover, the measured differential spectra containing the overall effects from arbitrarily oriented microcrystals are qualitatively well-described by a two-band model with a cosine band structure. By considering different orientations of the microcrystals in our simulations, we demonstrate that the contribution from the direction with the largest periodicity, i.e., the $\overline{\Gamma Z}$ direction $c = 12.5$ Å, strongly dominates the transient changes of the optical response. The large unit cell and the small bandwidths along one direction of this material allows for optical switching with up to ~40% transmission modulation depth using relatively moderate biasing fields. Also, the optical modulation of the material is extremely fast (sub-20 fs), as demonstrated directly by the quasi-instantaneous response to an electric field oscillating at mid-infrared (IR) frequency. These findings, together with its renowned characteristics, make MAPbI$_3$ a strong candidate for cost-effective, efficient, fast, and sensitive optical modulator materials.

## Results and discussion

**Experimental observation of Wannier–Stark localization.** For applying the strong transient bias, we employ phase-stable multi-cycle optical pulses with a center frequency of 20 THz (Supplementary Fig. 1). This frequency is non-resonant in energy with any of the optical phonons and electronic transitions, as the MAPbI$_3$ perovskite has a direct bandgap of $\varepsilon_{gap} = 1.62$ eV (390 THz, Fig. 1c) at room temperature and optical phonon modes below 10 THz (mainly from Pb-I inorganic sublattice) and above 26 THz (from methylammonium organic molecular vibrations)[26]. The phase-stable THz-biasing fields are generated using a difference-frequency generation scheme in GaSe[22,23] and are characterized by ultra-broadband electro-optic sampling[27]. The accuracy for determination of the absolute electric-field strength in the center of the pump spot is estimated to be ±15%. The peak field strength at the interior of the MAPbI$_3$ perovskite sample is obtained using the Fresnel transmission coefficient with the reported refractive index $n = 2.2$[28] at near-IR frequency, which sets the upper limit of the actual field strength. The sample is a polycrystalline film with a thickness of ~300 nm spin coated[29,30] on a cyclic olefin/ethylene copolymer substrate (TOPAS®, Supplementary Fig. 2)[31]. Due to the presence of the

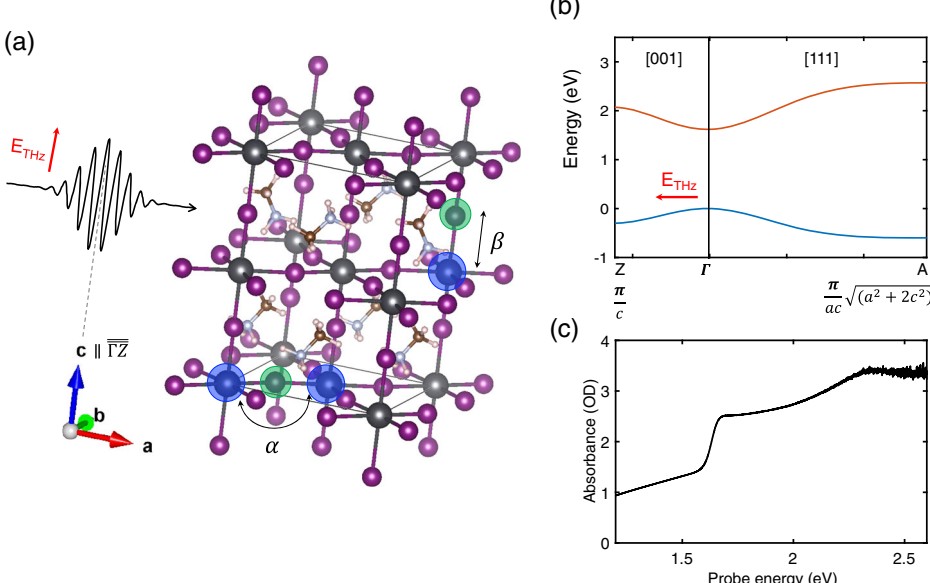

**Fig. 1 Experimental scheme and properties of MAPbI$_3$ perovskite. a** THz pulse geometry with a tetragonal unit cell (black rectangular cuboid) of MAPbI$_3$ (dark gray: Pb, purple: I, brown: C, light blue: N, light pink: H). The THz biasing along the c axis (parallel to the $\overline{\Gamma Z}$ direction) of a crystallite is depicted. The Pb-I-Pb angle along the diagonal direction of the a and b axis is denoted as $\alpha$, and the Pb-I bond lengths along the c axis is denoted as $\beta$. **b** Simplified electronic band structure of MAPbI$_3$ in the tetragonal phase along the directions $\Gamma(0,0,0) \rightarrow Z(0,0,0.5)$ and $\Gamma(0,0,0) \rightarrow A(0.5,0.5,0.5)$. The bandwidths and the lattice parameters are used from ref. [24]. **c** Optical absorption spectrum of MAPbI$_3$ in the spectral range of the probe pulses.

organic cation with a low rotational barrier[32], the crystal shows some degree of disorder at elevated temperature and a less pronounced periodicity compared to all-inorganic perovskites[32,33]. The differential transmission induced by the external electric-field transient is probed by near-IR and visible probe pulses, with spectra covering broad interband electronic transition energies between 1.4 and 2.4 eV (see Supplementary Fig. 3). The duration of these probe laser pulses is 7 fs, which is significantly shorter than the half-cycle period of the THz pump transients of 25 fs. Details of the experimental settings are described in the "Methods" section and ref. [21].

Figure 2a shows the differential transmission $\Delta T/T$ upon applying the THz biasing as a function of delay time between the pump and probe pulses. The peak field strength of the THz pump pulses is 6 MV/cm. As expected for the non-resonant THz pulse, the optical response of the material is instantaneous and peaks when the THz field strength is maximal. The modulation occurs at ~40 THz, i.e., twice the frequency of the THz pulse (Fig. 2b and Supplementary Fig. 4), as the measured differential transmission is at least a third-order nonlinear process[34]. In such a centrosymmetric crystal as the room-temperature tetragonal phase of perovskite MAPbI$_3$[35], no contribution from the electro-optic effect is expected, which is linear in the electric bias field. The clear temporal modulation of differential transmission appears at high fields, $-100 < \tau < 100$ fs, as the strong $E$ field shortens the interband dephasing time in the vicinity of the bandgap to be comparable to the half-cycle period of 25 fs of the THz transient. Thus, the precise arrival time of the probe pulse exciting the interband polarization was resolved within the dephasing time. It is noteworthy that the bandwidth of the THz pulse is ~4 THz (<40 THz modulation, Supplementary Fig. 4), so that in principle impulsive stimulated Raman excitation of sub-4 THz modes is possible. However, no oscillatory signal was observed after 150 fs, which is much shorter than the dephasing times of reported phonon modes with frequency up to 4 THz[29,36]. Therefore, any possibility of coherent phonon contribution to the temporal modulation can be ruled out.

More importantly, two distinct regimes can be identified in the time-resolved transient spectrum (Fig. 2a). The first regime appears at delay times $\tau < -100$ fs, where the field strength is relatively weak ($E < 3$ MV/cm), as an induced absorption (blue, $\Delta T/T < 0$) right below and an induced transmission (red, $\Delta T/T > 0$) right above the bandgap of $\varepsilon_{gap} = 1.62$ eV. The second regime is apparent for field strengths $E > 3$ MV/cm, occurring between delay times $-100 < \tau < 100$ fs (Fig. 2b). The transient response covers a significantly extended spectral range, compared to the moderate field regime. The induced transmission (red) above the bandgap now reaches up to $\varepsilon_{pr} = 1.9$ eV, where it abruptly switches to induced absorption (blue, $\Delta T/T < 0$). This negative region of $\Delta T/T < 0$ persists at probe energies all the way up to $\varepsilon_{pr} = 2.4$ eV. Also, the maximum modulation depth becomes as large as 38% at the probe energy of $\varepsilon_{pr} = 1.7$ eV (Fig. 2a and Supplementary Figs. 4 and 5).

The extended structure of the transient spectral response can be understood with the assistance of Fig. 2c, d. The localized Wannier–Stark states, equally spaced in energy by an amount $eE_{THz}D$, are depicted in the real-space along the field direction, $z$, in Fig. 2c. $D$ is the lattice period unit length and $n$ the index. This space-dependent energy shift results in differentiating the electronic transition energies within the same site (arrow with $n = 0$, Fig. 2c) from between different sites (arrows with $n = \pm1$, Fig. 2c). As the difference in the transition energy with respect to the central spatially direct ($n = 0$) transition is $neE_{THz}D$, one could assign the induced absorption below the bandgap and above 1.9 eV to be $n = -1$ and $n = 0$ transitions, respectively (Fig. 2d). The reduced absorption right above the bandgap stems from the spectral transfer from non-perturbed optical transition to red- ($n = -1$) and blue- ($n = 0$) shifted transitions (Fig. 2d). Depending on the strength of $E_{THz}$ and the degree of localization, $|n| > 1$ transitions could, in principle, also be observed. In this case (Fig. 2a), the observed single central step from reduced to increased absorption near the center of the band $\varepsilon_{pr} = 1.9$ eV is a noticeable signature of Stark localization, where the Wannier–Stark states are localized onto one unit cell.

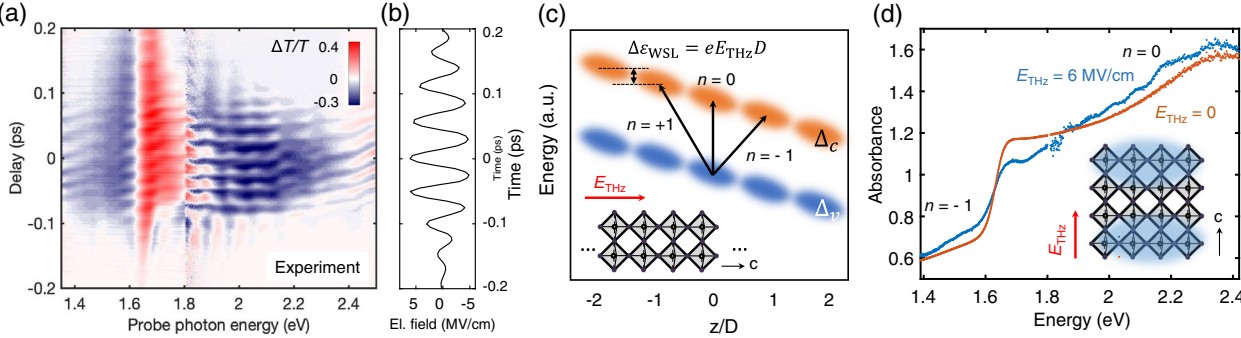

**Fig. 2 Experimental observation of the transient Wannier–Stark localization and the visualized diagram. a** Experimental differential transmission spectra on a polycrystalline film of MAPbI$_3$ perovskite at room temperature, as a function of delay time of probe pulses after THz pump pulses. The THz pulses have a peak field strength of 6 MV/cm and a center frequency of 20 THz; the probe pulses have photon energy of 1.4 ~ 2.4 eV. **b** Temporal profile of the applied THz bias transient. **c** Schematic picture of Wannier–Stark localization. In the presence of strong external fields along the $c$ axis, electronic states (orange: conduction band, blue: valence band) are localized to a few layers of $ab$ plane and energetically separated by $\Delta\varepsilon_{WSL} = eE_{THz}D$ between adjacent lattice sites. Black arrows depict the interband transitions within the same site ($n = 0$) and between different sites ($n = \pm 1$). **d** The absorbance with and without the external transient biasing. The Wannier–Stark localization effectively reduces the 3D electronic structure into 2D layered structure along the $ab$ plane, as depicted in blue together with the simplified 3D structure. In case of $E_{THz} = 6$ MV/cm in considering the lattice constant $D$ of 12.5 Å, $\Delta\varepsilon_{WSL} = eE_{THz}D$ is estimated to be 750 meV, consistent with the spectrum showing that the absorption band of $n = -1$ and $n = 0$ are ~750 meV apart.

It is important to distinguish this transient Wannier–Stark localization from the optical Stark-type effects such as the Autler–Townes effect[37] and the Bloch–Siegert shift[38]. In general, an external electric field affects the optical properties of a semiconductor in two ways: there are spectral and kinetic aspects[39]. Spectral aspects refer to energy shifts and broadenings that arise from the mixing of two states by the external optical field. The mixing of the wavefunctions results in dressed states and leads to the Stark-type shifts. The magnitude of such shifts increases with the amplitude of the incident field and the interband dipole matrix, but decreases with increasing detuning between the light frequency and the transition frequency. On the other hand, kinetic aspects represent the evolution of the particle distributions in the renormalized states driven by the external field, which is called "intraband acceleration." This intraband acceleration leads to the Franz–Keldysh effect at a moderate field strength and eventually Wannier–Stark localization in the strong-field regime.

Each of these two contributions can be straightforwardly separated in the semiconductor Bloch equations[40–42]. Specifically, the third and second terms on right-hand side of the Eq. (1) represent the spectral (optical Stark-type effects) and the kinetic (Wannier–Stark localization) aspects, respectively. One can thus directly compare each contribution to the differential optical response. As evident from the Supplementary Fig. 6, the THz-induced optical Stark effect is shown to be much weaker (on the order of a few meV) compared to the shifts arising from the Wannier–Stark localization, which corresponds to approximately half the bandwidth (several 100 meV). Therefore, we conclude that the observed transient response is mainly contributed from the Wannier–Stark localization.

Furthermore, to analyze the possible effects of THz-induced generation of virtual and real carriers that may arise from multi-photon transitions, we also solve a full set of semiconductor Bloch equations in which the THz field is included non-perturbatively and the weak optical probe pulse is considered linearly (Supplementary Methods). The results (Supplementary Fig. 6) show that for the considered field amplitudes, such higher-order interband effects arising from the THz fields are negligible, as the results from the full equations are very close to the ones obtained from the simplified Eq. (1). Finally, the contribution from the possible higher energy bands within our probe photon energy range[43] is negligible. Indeed, we neither observe any additional Franz–Keldysh and/or Wannier–Stark response within our probe energy, nor find any decay of the entire signal as a function of the field strength due to the tunneling to higher energy bands at intense field regime. Therefore, we could consider simple two-band systems to understand our experimental demonstration of Wannier–Stark localization in further detail. As will be shown below, the two-band model explains our observations.

By driving the three-dimensional (3D) system into Wannier–Stark localization, i.e., localizing it in the field direction, we transiently create an effectively two-dimensional (2D) electronic system (Fig. 2c, d). Given the unit cell doubling, this optically prepared transient 2D system perpendicular to the $c$ axis may be directly compared to the physically isolated double-layer structure of PbI$_6$ octahedra. In such 2D perovskites as (BA)$_2$(MA)$_{l-1}$Pb$_l$I$_{3l+1}$ perovskites[44], the inorganic layers (perpendicular to the $c$ axis in 3D equivalence) are separated by bulky organic layers[45]. The bandgap of the 2D quantum well perovskites is widened due to the bandwidth narrowing (mainly due to the zero dispersion along the vertical direction) compared to 3D perovskite[46]. In the case of (BA)$_2$(MA)$_{l-1}$Pb$_l$I$_{3l+1}$ perovskites, where the PbI$_6$ octahedral network forms a double layer ($l = 2$), the same periodicity of the sample along the $c$ axis, the optical bandgap is ~2.1 eV, which is comparable to the observed 1.9 eV[44]. It is noteworthy that the observed Wannier–Stark step at $\varepsilon_{pr} = 1.9$ eV under THz fields is slightly lower than the expected value under static fields due to the spectral broadening induced by the THz modulation, as will be discussed below. Therefore, the abrupt shift of the absorption edge from $\varepsilon_{pr} = 1.6$–1.9 eV at high transient fields (Fig. 2d) could be attributed to the transfer of spectral weight from $\alpha(\varepsilon_{g,3D} < \varepsilon_{pr} < \varepsilon_{g,2D})$ to $\alpha(\varepsilon_{g,2D} < \varepsilon_{pr})$. Such a THz-induced reduction of dimensionality from a 3D to a 2D system could enable new applications in both transport and optoelectronics due to the relatively easy access to that regime in these hybrid perovskite materials.

**Simulations considering one orientation**. To capture the essential ingredients responsible for the experimental observations, we carry out theoretical calculations based on different models of increasing complexity. We start with considering perfect alignment of the THz field with the direction along which the joint bandwidth of the highest valence and the lowest

conduction band is narrowest. For the case of the tetragonal MAPbI$_3$ perovskite, the narrowest joint bandwidth, $\Delta_{\overline{\Gamma Z}} = 0.75$ eV, is along the $\overline{\Gamma Z}$ direction (Fig. 1b)[24]. We thus take into account two 1D bands, i.e., one valence and one conduction band with a cosine-like (tight-binding) band structure and the bandgap of 1.62 eV. Thus, the energy difference for interband transitions is taken as $\varepsilon_{cv}(k) = 1.62$ eV $+ (\Delta_{\overline{\Gamma Z}}/2)(1 - \cos(g(k, a^*) ka^*))$ (see "Methods" section for details of the function $g(k,a^*)$). For this model, the spectra are obtained by numerically solving the semiconductor Bloch equations[40–42], as described in the "Methods" section.

Already when considering static fields (Fig. 3a), the simulation results obtained by this simple model exhibits substantial qualitative similarities with the transient experimental results shown in Fig. 2a (c.f. for the case of $\overline{\Gamma A}$ direction, Supplementary Fig. 7). For all field strengths, increased absorption is present below the bandgap and reduced absorption directly above the bandgap. For rather weak field strengths of up to about 0.5 MV/cm, oscillations arising from the Franz–Keldysh effect are visible, shifting towards the band center with increasing field. For fields exceeding ∼3 MV/cm, signatures of Wannier–Stark localization become noticeable, as the field-dependent interband transition energies shift to higher and lower energies by $neED$ with increasing $E$ (Fig. 2c). Starting at around 3 MV/cm, the condition for Stark localization is fulfilled, i.e., $eED > \Delta/2$ (meaning that the energy of the ($n = -1$) Wannier–Stark state is in the bandgap region, see Fig. 2c, d) and, therefore, the dominant feature is the step-like change from reduced absorption to induced absorption in the center of the band at 1.97 eV (this value is the average transition frequency within our model). This step-like change is, in fact, also the main feature visible in the experimental results for sufficiently high fields, i.e., between about $-100 < \tau < 100$ fs as shown in Fig. 2a.

Besides, by considering pulsed THz fields, the simulated differential spectra with the same model (Fig. 3b, c) well describe both spectral and temporal features in the observed transient modulation of differential transmission spectra (Fig. 2a). Figure 3b shows the negative change of the transient absorption, $-\Delta\alpha_{\overline{\Gamma Z}}$, upon non-resonant biasing with a THz pulse with a peak field strength of $E_0 = 6$ MV/cm and a center frequency of 20 THz, as shown in Fig. 3c. Besides temporal modulation of the entire transient spectra at twice the carrier frequency of the THz transient, the dominant feature at sufficiently large field strengths ($-100 < \tau < 100$ fs) is the rapid change from increased to reduced

transmission in the center of the band $\varepsilon_{pr} = 2$ eV, which originates from Stark localization. The slightly lower value of the observed central step at $\varepsilon_{pr} = 1.9$ eV and the asymmetric nature of the spectral shape with respect to the central step (Fig. 2a) compared to this simplified model (Fig. 3b) can be explained by the polycrystallinity of the system as discussed below. Given the complexity, disorder, and polycrystallinity of the investigated sample, the required field strength at which this step starts to appear is in surprisingly good agreement with the experiment, which confirms that the observed response constitutes a clear sign of Wannier–Stark localization. Our interpretations are further supported by Supplementary Fig. 7, which shows how the results of Fig. 3 change if we consider that the THz field is aligned with the $\overline{\Gamma A}$ direction instead of the $\overline{\Gamma Z}$ direction. Comparing those two figures clearly shows that due to the larger bandwidth in the $\overline{\Gamma A}$ direction, the Wannier–Stark localization requires higher field amplitudes to develop and, furthermore, would lead to a transition from reduced to induced absorption at significantly higher energies as observed in the experiment. The effects of different field directions and the averaging over them is discussed in more detail below (see Fig. 4).

As demonstrated so far, Wannier–Stark localization starts to occur at the field amplitude as low as 3 MV/cm in the MAPBI$_3$ perovskite, due to the relatively large periodicity, the narrow joint bandwidth, and the coincidence of the two along the same direction. The largest lattice constant of tetragonal MAPBI$_3$ perovskite, along the $c$ axis, $c = 12.5$ Å, is more than twice as large as those of conventional all-inorganic semiconductors crystallizing with strong covalent bonds in the diamond, wurtzite, or zincblende forms (3.5 ∼ 6.5 Å at 300 K). This finding arises because (i) the cubic perovskite unit cell is expanded through rotation of $ab$ plane by 45° and cell doubling along the $c$ axis in the tetragonal phase; and (ii) the pseudocubic lattice parameter formed by relatively large $Pb^{2+}$ and $I^-$ ions is 6.3 Å[25], which is at the larger side of the distribution of parameters for cubic lattice parameters. The pseudocubic lattice parameter is large enough to accommodate large organic molecular cations within the void of their network.

The direction of the narrowest joint bandwidth of the conduction and valence bands, $\overline{\Gamma Z}$, coincides with the $c$ axis (Fig. 1a). The conduction band is composed of the overlap of $Pb(6p)$-$I(5p)$ atomic orbitals and the valence band is of that of $Pb(6s)$-$I(5p)$ orbitals[7]. Thus, the Pb-I bond length as well as the largest Pb-I-Pb angle could determine the widths of both bands

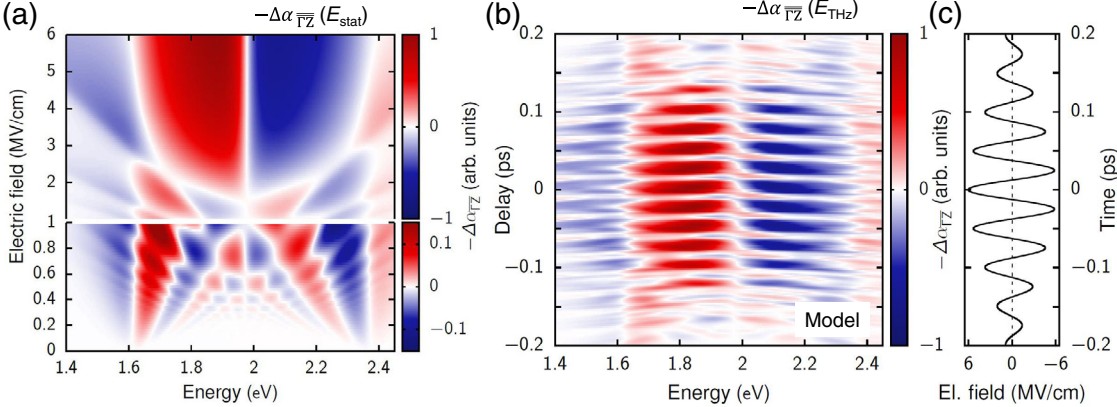

**Fig. 3 Numerical simulation of differential absorption spectra. a** Negative change of the optical interband absorption $-\Delta\alpha_{\overline{\Gamma Z}}$ for static fields from a cosine band modeling along $\overline{\Gamma Z}$ direction. The region of electric-field strengths up to 1 MV/cm is enlarged to show Franz–Keldysh oscillations and the transition to the Wannier–Stark regime. **b** Calculated $-\Delta\alpha_{\overline{\Gamma Z}}$ spectra for the excitation with a THz pulse with a peak field strength of $E_0 = 6$ MV/cm, where the delay $\tau$ between the THz and the optical pulse is varied. **c** Simulated temporal profile of the applied THz bias transient. The pulse duration $\overline{T}$ is 240 fs, the THz frequency is 20 THz, and the dephasing time is $T_2 = 20$ fs.

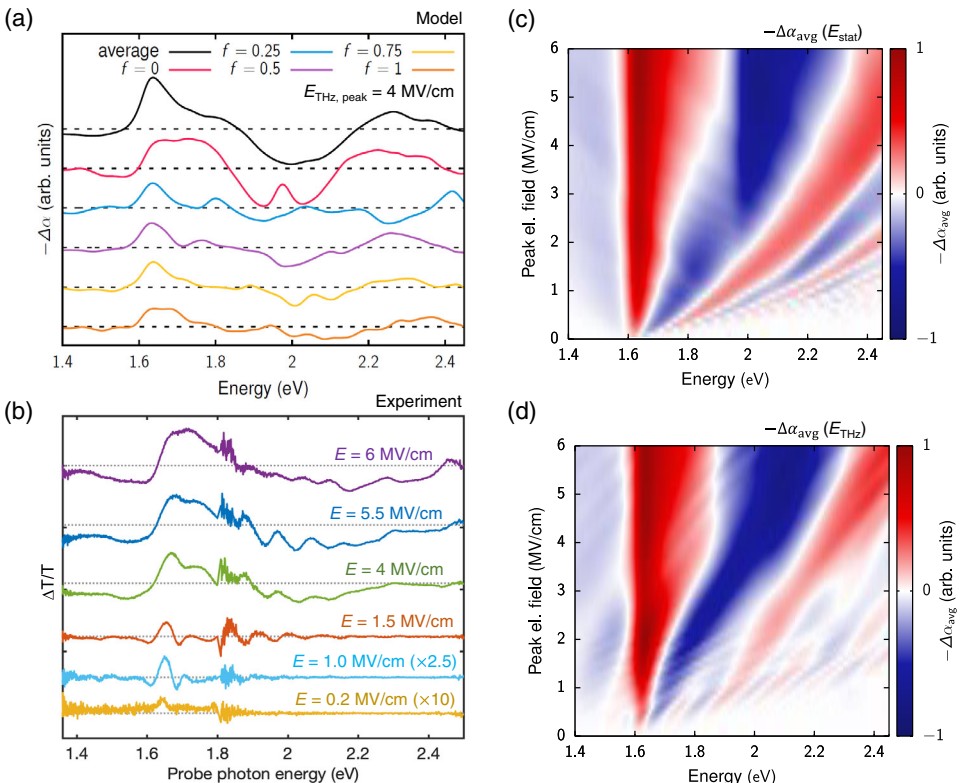

**Fig. 4 Experiments on polycrystalline system and simulations with averaging of cosine band model from ΓZ to ΓZ direction. a** Illustration for the averaging process over the interpolation parameter $f$ from the $\overline{\Gamma Z}$ direction ($f = 0$) to $\overline{\Gamma A}$ direction ($f = 1$). The negative absorption changes $-\Delta\alpha_f$ are calculated for different one-dimensional systems using a THz pulse centered at $t = 0$, with an amplitude of $E_0 = 4$ MV/cm, a pulse duration of $\overline{T} = 240$ fs, and a THz center frequency of 20 THz. **b** Temporal slices of $\Delta T/T$ as a function of probe photon energy (Fig. 2a), at a delay time corresponding to the contour with constant electric-field amplitudes $E$ (Fig. 2b). **c** Averaged absorption change, $-\Delta\alpha_{avg}$, for static fields of various strengths. **d** Averaged absorption change, $-\Delta\alpha_{avg}$, for a THz pulse centered at $t = 0$ and various field strengths.

and the magnitude of the bandgap. In the tetragonal MAPbI₃ perovskite, the corner-shared PbI₆ octahedra in cubic phase are tilted about the $c$ axis in the opposite direction between successive tilts, which reduces the Pb-I-Pb angle (Fig. 1a, denoted as $\alpha$) from 180° along the diagonal direction of the $a$ and $b$ axis. The smaller Pb-I-Pb bond angle indicates weaker orbital overlap between Pb and I atoms, and thus smaller band dispersion along $\overline{\Gamma M}$ than $\overline{\Gamma Z}$. However, the Pb-I bond lengths along the $c$ axis (Fig. 1a, denoted as $\beta$) is known to be longer on average[47] and has greater effect on the dispersion than the angle due to the $\sigma$ bonding nature, which leads to the coincidence of the direction of the largest lattice constant and the narrowest bandwidth. We note that unlike GaAs, the body diagonal direction exhibits the strongest dispersion ($\overline{\Gamma A}$). Overall, the large ionic diameter and the geometric distortion result in the unusually narrow joint bandwidth, lower than 1 eV.

**Including polycrystallinity by averaging over orientations.** We now account for the system's polycrystallinity by considering contributions to the differential transmittance spectra from crystallites with orientations different from those with the $c$ axis parallel to the THz field polarization. To include arbitrary orientations of the crystallites into our simulations, we take the $\overline{\Gamma Z}$ and the $\overline{\Gamma A}$ directions, i.e., the two extreme directions with the narrowest/broadest bandwidth and simultaneously the smallest/ largest distance in $k$-space (see Fig. 1b), and perform an average over all in between bandwidths and extensions of the first Brillouin zone (see "Methods" section and Supplementary Figs. 8 and 9), by interpolating between the two limiting cases with a

parameter $f$. The simulated absorption changes at a field amplitude of $E_0 = 4$ MV/cm with various interpolation parameters $f$ are shown in Fig. 4a together with the measured differential spectra at different instantaneous field amplitudes of the THz pulse (Fig. 4b). Here, $f = 0$ denotes the response along the $\overline{\Gamma Z}$ direction (i.e., the $c$ axis) and $f = 1$ along the $\overline{\Gamma A}$ direction.

As shown in Fig. 4a, the absorption changes depend strongly on the interpolation parameter $f$, i.e., on the bandwidth and the distance to the border of the first Brillouin zone. For $f = 0$, which corresponds to the $\overline{\Gamma Z}$ direction, the field amplitude of $E_0 = 4$ MV/cm drives the system into the region of Stark localization. Therefore, for a static field of such an amplitude, one would see a strong induced absorption in the band center at 1.97 eV, which corresponds to an optical transition to the Stark localized state. The transient nature of the THz pulse causes the single negative peak to be split into two peaks and the spectral region of induced absorption to be slightly broadened. With increasing $f$, both the bandwidth and the distance to the border of the first Brillouin zone increase. As a result, the minimum field strength for which Stark localization is realized increases significantly by approximately a factor of $(c/a^*_{\overline{\Gamma Z}})(\Delta_{\overline{\Gamma A}}/\Delta_{\overline{\Gamma Z}})$, equaling about 4.7. Consequently, already for $f = 0.25$, the absorption changes show no sign of Stark localization, with several oscillations emerging, owing to the THz driving. This trend of overall weaker absorption changes with some oscillatory structure is also present for even larger $f$. The only feature present in all spectra shown in Fig. 4a is some induced absorption below the bandgap and reduced absorption directly above the bandgap.

However, when averaging over the interpolation parameter $f$, i.e., over the orientations considered by our modeling, the result reproduces the main features present for $f = 0$, with somewhat fewer oscillations (black curve in Fig. 4a). The averaged graph is in good agreement with the differential spectra at high field amplitudes given the very few parameters used to describe the entire contributions from the complex actual band structure (i.e., two extreme bandwidths, lattice constants, and linear interpolation of them). The prediction of 1.9 eV for change from bleaching to induced absorption as is observed in the experiment (Fig. 4b, upper curves) is remarkable. A few minor features including the shape at higher photon energy could be improved by modifying the model band structure and the oscillations around 2 eV in Fig. 4b need further studies (Supplementary Fig. 10). More importantly, this position of 1.9 eV is very close to the center of the interband absorption for the $\overline{\Gamma Z}$ direction, 2 eV as the single direction model indicated above. Thus, in the averaged results, the spectra for small $f$ dominate strongly, as (i) the absorption changes are spectrally concentrated in the monitored region due to the small bandwidth, (ii) one is in the regime of Stark localization due to the small extent of the first Brillouin zone, and (iii) for larger $f$ the rather weak and oscillatory results partly cancel each other. For these reasons, the contribution from the $\overline{\Gamma Z}$ direction, corresponding to small $f$, is enhanced for energies far above the bandgap and dominates the entire phenomenon.

The results of Fig. 4a, b suggest that, for the randomly oriented crystallites in the film, the overall response is dominated by the response originating from the band dispersion in the $\overline{\Gamma Z}$ direction. This reasoning is substantiated by the averaged field-dependent absorption changes calculated for both a static and a THz field shown in Fig. 4c, d, respectively. This finding is remarkable, because it means the two extreme cases—the completely random orientation of a polycrystalline sample and the perfectly oriented single crystal—are expected to produce very similar optical responses. The only slight difference between the two extremes would be a small shift in the photon energy (~100 meV) where the induced transmission turns to the induced absorption and in the transient spectral shape. The two extremes include partial preferential orientations. We also note that the averaging process using only two extreme directions does not contain any material-specific information, which means that one could expect other polycrystalline materials to behave similarly.

As expected, the $\overline{\Gamma Z}$ direction dominates the averaged results, which include the contributions from the dispersion in all the other directions. In both cases for strong fields, the dominant feature is a rapid change from reduced to increased absorption, which takes place near the center of the interband absorption that corresponds to the dispersion in the $\overline{\Gamma Z}$ direction. Due to the spectral broadening induced by the THz modulation, this transition appears at slightly lower photon energies for the THz field (Fig. 4c) than for the static field (Fig. 4d). Thus, Fig. 4c, d is consistent with the notion that the step-like sign change in the center of the band for sufficiently strong field amplitudes is a signature of Stark localization for the polycrystalline perovskite sample.

In conclusion, we have demonstrated that solution-processed, polycrystalline MAPbI$_3$ shows optical transmission change by tens of percent at relatively modest field strengths via transient Wannier–Stark localization. The large lattice periodicity, the narrow electronic energy bandwidths, and the coincidence of these two along the same high-symmetry direction promotes this material to the Wannier–Stark regime under relatively moderate biasing fields. Polycrystallinity of this material turns out not to hinder the Wannier–Stark localization effect as observed, due to the dominant contribution from the least dispersive direction of the band structure, which favors low-cost fabrications with this material as optical modulators. The degree of disorder and relative orientation among crystallites may influence the modulation spectral shape slightly, e.g., the position of the photon energy where the induced transmission to induced absorption happens, which could be finely tuned depending on the desired device performance by further systematic studies.

Moreover, the phase-stable THz field transients and the ultra-broadband optical pulses of 7 fs duration revealed that the optical modulation of this material has an extremely fast, quasi-instantaneous (sub-20 fs) temporal response in visible/near-IR spectral region. This technique could be generalized for realizing transient Wannier–Stark localization in other semiconductor solids in a carefully prepared single-crystalline or a polycrystalline form. More generally, this method enables to analyze any ultrafast changes in optical properties induced by the phase-locked and intense electromagnetic field transients, be it resonantly or non-resonantly. Although here we used only the electric field of the transients, one could also exploit the magnetic component for exploring ultrafast magneto-optic effects, by enhancing the magnetic field with respect to the electric field with, e.g., a specially designed plasmonic nanoaperture.

Finally, instead of semiconductor superlattices, which need expensive high-vacuum manufacturing processes, the solution-processed hybrid perovskites could meet the growing need for cost-effective[2], efficient, fast, and sensitive characteristics as optical modulators[48]. Together with the renowned photophysical properties of MAPbI$_3$, such as the long carrier diffusion length[4,5], low mid-gap trap density[5,7], and large absorption coefficient[3], this finding of high modulation depth, ultrafast response, and low-onset field for Wannier–Stark localization highlights the potential of this material in photonic applications[49,50].

## Methods

**Experimental details**. The phase-stable multi-cycle THz pulses with a peak field strength of ~10 MV/cm are generated using difference-frequency mixing (DFG) in GaSe[22,23]. The regeneratively amplified pulses with 780 nm and 130 fs are used to pump two parallel optical parametric amplifier stages to provide tunable near-IR pulses with minimum relative phase fluctuation. The two near-IR pulses are then combined and sent to the GaSe nonlinear crystal for the DFG. The thus generated THz pulses are focused onto the sample with off-axis parabolic mirrors of focal length $\tilde{f} = 15$ mm and effective NA = 0.2. The electric-field transient is characterized by ultra-broadband electro-optic sampling[27] at a 30 μm-thick GaSe crystal using balanced detection of an 8 fs probe pulse centered at a wavelength of 1.2 μm as the gating pulse. The quantitative value of the field amplitude is obtained by measuring the THz average power, pulse repetition rate, and focal spot size. Then, the value at the interior of the MAPbI$_3$ perovskite sample is estimated using the Fresnel transmission coefficient for the THz field at the air–MAPbI$_3$ interface.

For detection of the field-induced differential optical transmittance in broad spectral range, we generate near-IR and visible pulses with the duration of 7 fs by non-collinear optical parametric amplification (Supplementary Fig. 1)[51]. The probe pulses are combined with the mid-IR pump pulses at a germanium beam splitter so that both pulses co-propagate through the sample. The probe pulses are then dispersed onto a spectrometer coupled to a charge-coupled device (CCD) camera for the spectral resolution. The relative timing between the pump and probe pulses was controlled using an optical delay stage. To detect the differential optical transmission spectra, we modulate the mid-IR pump pulses by an optical chopper operating at 125 Hz, which is synchronized with the 1 kHz laser repetition rate and the readout of the CCD camera. Two subsequent spectra taken from the CCD camera are subtracted by each other and normalized by one spectrum without the pump. The sample compartment in the experimental setup was purged with dry nitrogen in order to avoid degradation. The complete experimental setup and the laser system have been fully illustrated in ref. [21].

**Theoretical approach**. For calculating the linear-optical interband absorption spectra, we numerically solve the semiconductor Bloch equations, including the intraband acceleration induced by the strong THz field[40–42]. We use a 1D trajectory in $k$-space, denoted as the $\overline{\Gamma x}$ direction where $x$ is an arbitrary point in the 1. Brillouin zone, which is parallel to the polarization direction of the incident THz field and goes through the Γ-point of the Brillouin zone. In the linear optical regime, the semiconductor Bloch equations reduce to the equations of motion for

the microscopic polarizations $p_k^{cv}$ and read

$$\frac{\partial}{\partial t}p_k^{cv} = \frac{i}{\hbar}\varepsilon_{cv}(k)p_k^{cv} + \frac{e}{\hbar}E_{THz}(t)\nabla_k p_k^{cv} - \frac{i}{\hbar}E_{opt}(t)\mu_k^{vc} - \frac{p_k^{cv}}{T_2} \qquad (1)$$

Dephasing processes are treated phenomenologically by adding the dephasing time $T_2$.

For all calculations presented in this study, we include the intraband dynamics induced by the static or pulsed THz fields to infinite order, whereas the weak optical probe of the interband absorption is considered only to the first order. In this linear-optical regime, we thus neglect carrier generation by multi-photon processes and impact ionization, which does not seem to play a dominant role in the measured transient spectra. Interband tunneling by the THz field could lead to bleaching at later delay times and the slightly asymmetric spectral evolution with respect to $\tau = 0$ (Fig. 2a) (corresponding to the trailing edge of the THz transient in the Supplementary Material of ref. [21]). However, significant carrier multiplication does not occur within this experimental window, as shown in Supplementary Fig. 11.

For the interband dipole matrix element, we use the usual decay with increasing transition frequency[39]

$$\mu_k^{vc} = \mu_0 \frac{1.62\,\text{eV}}{\varepsilon_{cv}(k)} \qquad (2)$$

where the choice of $\mu_0$ is not relevant here, as it contributes only as a prefactor to the absorption spectra.

For the THz pulses, we use a Gaussian envelope

$$E_{THz}(t) = E_0 e^{-4\ln(2)(\frac{t-\tau}{\bar{T}})^2} \cos(\omega_{THz}(t-\tau)) \qquad (3)$$

with the electric-field amplitude $E_0$, the pulse duration $\bar{T}$ (FWHM of the intensity), the time delay $\tau$, and the THz frequency $\omega_{THz}$. The optical probe pulse is modeled as a weak ultrashort delta-like pulse.

The total optical polarization is obtained by summing over the microscopic polarizations $p_k^{cv}$

$$P(t) = \sum_k (\mu_k^{vc}) * p_k^{cv}(t) + c.c. \qquad (4)$$

By Fourier transforming the macroscopic polarization $P(t)$, the linear absorption can be obtained by

$$\alpha_{1D,\overline{\Gamma x}}(\omega) \propto \omega\,\text{Im}(P(\omega)) \qquad (5)$$

To be able to compare the numerical results for the 1D $k$-space trajectory to the measured $\Delta T/T$ spectra, the negative change of the optical absorption in three dimensions $-\Delta\alpha_{3D}$ is calculated assuming a parabolic electronic dispersion perpendicular to the considered 1D direction. Due to the constant two-dimensional density of states for a parabolic dispersion, the absorption of the corresponding three-dimensional system is easily obtained as ref. [21]

$$\alpha_{\overline{\Gamma x}}(\omega) \propto \int_0^\omega \alpha_{1D,\overline{\Gamma x}}(\omega')d\omega' \qquad (6)$$

**Band structure model and averaging over crystallographic directions**. To incorporate both the bandwidth and the effective mass $m^*$ at the bandgap as obtained from ab initio calculation in ref. [24] into our model, we use an interband energy difference of

$$\varepsilon_{cv}(k) = \varepsilon_{gap} + \frac{\Delta}{2}(1 - \cos(g(ka^*)ka^*)) \qquad (7)$$

Here, $\pi/a^*$ is the distance from the $\Gamma$-point to the border of the first Brillouin zone and the interpolation function

$$g(ka^*) = f + (1-f)\frac{ka^*}{\pi} \qquad (8)$$

guarantees that $\varepsilon_{cv}(0) = \varepsilon_{gap}$ and $\varepsilon_{cv}(\pm\pi/a^*) = \varepsilon_{gap} + \Delta$, meaning the bandgap energy $\varepsilon_{gap}$ and the bandwidth $\Delta$ are preserved.

The parameter $f$ is adjusted to obtain the effective mass which corresponds to the second derivative of the band structure at the $\Gamma$ point:

$$m^* = \hbar^2 \left[\frac{d^2\varepsilon_{cv}(k)}{dk^2}\bigg|_0\right]^{-1} \qquad (9)$$

as given in ref. [24].

As mentioned before, the polycrystallinity of the system is included by averaging over several differential transmittance spectra.

The transition from the $\overline{\Gamma Z}$ to the $\overline{\Gamma A}$ direction is carried out by varying the bandwidth $\Delta$ from $\Delta_{\overline{\Gamma Z}} = 0.75$ eV to $\Delta_{\overline{\Gamma A}} = 1.55$ eV, the extent of the first Brillouin zone $\frac{\pi}{a^*}$ from $\frac{\pi}{a^*_{\overline{\Gamma Z}}} = \frac{\pi}{c} = \frac{\pi}{1.27}$ nm$^{-1}$ to $\frac{\pi}{a^*_{\overline{\Gamma A}}} = \frac{\pi}{ac}\sqrt{2c^2+a^2} = \frac{\pi}{0.56}$ nm$^{-1}$ and the effective mass $m^*$ from $m^*_{\overline{\Gamma Z}} = 0.17m_0$ to $m^*_{\overline{\Gamma A}} = 0.09m_0$ via a parameter $f$, which varies from 0 (i.e., the $\overline{\Gamma Z}$ direction) to 1 (i.e., the $\overline{\Gamma A}$ direction)[24]. The interpolation

is performed as:

$$\Delta(f) = \Delta_{\overline{\Gamma Z}} + f(\Delta_{\overline{\Gamma A}} - \Delta_{\overline{\Gamma Z}})$$

$$\frac{\pi}{a^*(f)} = \frac{\pi}{a^*_{\overline{\Gamma Z}}} + f\left(\frac{\pi}{a^*_{\overline{\Gamma A}}} - \frac{\pi}{a^*_{\overline{\Gamma Z}}}\right)$$

$$m^*(f) = m^*_{\overline{\Gamma Z}} + f(m^*_{\overline{\Gamma A}} - m^*_{\overline{\Gamma Z}}) \qquad (10)$$

where $f = 0$ describes the $\overline{\Gamma Z}$ direction and $f = 1$ the $\overline{\Gamma A}$ direction, respectively.

The above described averaging of several spectra for the discretized parameter $f$ is performed via evaluating

$$\alpha_{avg}(\omega) = \frac{1}{n}\sum_{f_i}\alpha_{f_i}(\omega), i \in [1,n] \qquad (11)$$

with the respective absorption $\alpha_{f=0} = \alpha_{1D,\overline{\Gamma Z}}$ and $\alpha_{f=1} = \alpha_{1D,\overline{\Gamma A}}$, where for convergence $n$ is typically chosen as 51.

## Data availability

The datasets generated and/or analyzed during the current study (experimental and theoretical) have been deposited in the Edmond database at: https://edmond.mpdl.mpg.de/imeji/collection/WNz2cpxosLMJGjj8.

## Code availability

The code used for simulations and analyzing the data in this study is available from the corresponding author upon reasonable request.

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

## Acknowledgements

We thank Keno Krewer and Johannes Hunger for helpful discussions. T.M. and D.B. acknowledge financial support from the Deutsche Forschungsgemeinschaft (DFG, German Research Foundation) through the Collaborative Research Center TRR 142 (project number 231447078, project A02). M.B. and H.K. thank the DFG for financial support through the Collaborative Research Center TRR 288 (project number 422213477, project B07), the European Union's Horizon 2020 research and innovation program under grant agreement number 658467, and the Max Planck Society for financial support. A.L. and J.B. acknowledge financial support from the European Research Council through ERC Advanced Grant 290876 (UltraPhase) and the Carl Zeiss Foundation through the fellowship program.

## Author contributions

A.L. and H.K. designed and J.B and H.K. performed the time-resolved THz-induced differential optical transmittance measurement. H.K. analyzed and interpreted the experimental data with the assistance of A.L and J.B. T.M. conceived and D.B. carried out theoretical studies and numerical calculations. H.K. and T.M. wrote the manuscript together with D.B., M.B. and A.L. All the authors contributed to finalizing the draft. The manuscript was written through contributions of all authors. All authors have given approval to the final version of the manuscript.

## Funding

## Competing interests

The authors declare no competing interests.
