## [Peer Review File · Nature Communications]

Low-field onset of Wannier-Stark localization in a polycrystalline hybrid organic inorganic perovskiteREVIEWER COMMENTS

Reviewer #1 (Remarks to the Author):

Berghoff et al report Wannier-Stark localization in a thin film of hybrid perovskite MAPbI₃ achieved via non-resonant excitation with a 20THz electric field. This is achieved in a self-assembled material and at low electric field intensities compared to previous results in GaAs. Given the potential applications in ultrafast optically controlled switching of material properties, the presented results are not only novel but also have a broad interest. The experiment involves ultrafast THz pump – optical probe spectroscopy and supported by extensive and well-crafted calculations, comparing the effect of static and transient electric fields and accounting for the effect of the sample polycrystallinity. This work represents an exciting contribution, and is suitable for publication in Nature Communications. A few minor points to be addressed are as follows:

1. It would be helpful to clearly note that in the data presented in the main manuscript (Fig.2(a)), the low-field regime refers only to the early delay times before the THz field peak. A separate measurement, with an overall lower THz field, could also be shown for comparison, providing a reference where no localization occurs.
2. The discussion of figure 2, which shows the main experimental finding of the paper and its interpretation, should be expanded. In particular, panel d is not sufficiently explained in the main text: how are the separate peaks appearing in the absorbance attributed to $n=0, +/-1$? The possibility of optically controlled switching between 3D materials and QWs is fascinating and could be explored more at length.
3. Since the low-field onset of Wannier-Stark localization is a key claim of this work, is it possible to provide a confidence interval to the estimation of the THz electric field amplitude? Can it be absolutely calibrated e.g. using a thin (smaller than the coherence length between the EOS probe and all relevant THz wavelengths) electro-optic crystal?
4. Considering the in-depth discussion of orientations in both real space and Brillouin space in pages 12-13, a diagram not only of MAPbI₃ structure in general but of the specific details being addressed would facilitate understanding (e.g. what Pb-I bond angles? Which direction is Γ M?)
5. Is there a specific reason to consider MAPbI₃ rather than other hybrid perovskites? Could one expect to easily replicate these results in other materials in the same family? Would a softer lattice, e.g. by selecting different cations or substituting a lighter metal for Pb, result in localization at even lower fields?

A few typos and mistakes in figure presentation should also be corrected:

- * In the section "Experimental observation of Wannier-Stark Localization", units are incorrect in the sentence: "relatively weak fields, $E < 3$ MV, for $\tau < -100$ fs". It should read MV/cm.
- * Is all the data presented (e.g. in fig. S2, S3) measured with a 6MV/cm THz field? Please specify so in captions.
- * The x-axis of figure S1 is labeled incorrectly, it should probably be in eV and not nm
- * Figures S4/S5 are not referred to in the main text

Reviewer #2 (Remarks to the Author):

This work provided an experimental observation of transient Wannier-Stark (WS) localization in a polycrystalline MAPbI₃ perovskite based on similar previous work performed in the GaAs/AlGaAs superlattice (Ref. 8 in this manuscript). As the authors emphasized, the main difference is that the field amplitude needed here (3MV/cm) is far less than that in GaAs/AlGaAs (which exceeds 10 MV/cm) due to the large relevant lattice constant, the small width of electronic energy bands, and the coincidence of these two along the same high-symmetry direction. The results and the method discussed here are interesting and the manuscript is well written. However, since similar work has been reported in literature and I did not find enough novelty of current work, I cannot accept the present paper for publication in high level journals like Nat Comm. The authors may want to show and discuss about the novelty and new highlights of their work should they try to submit to similar journals again. For instance:

- 1) Further analysis and emphasis of the peculiarity unique to MAPbI₃ (such as low field strengths required for WS), and in particular the applications of this feature.
- 2) Any other typical physical phenomena not limited to WS localization that are based on the

transient spectral analysis? The authors developed a direct method to show WS localization in real natural solids. Since such localization has already been extensively studied in previous work, it will be interesting if this method can also be used for the study of other physical phenomena.

3) MAPbI₃ perovskite exhibits wide light absorption range and excellent photo-electronic properties and is considered as a prominent light harvester. In this manuscript, the description and discussion of novel properties of MAPbI₃, especially related to this work, seems absent.

Other questions and suggestions:

1. In this work, the WS localization is created by applying the bias field from the phase-stable THz pulse and can be detected by optical absorption spectra. Is this method generally applicable for single-crystalline and polycrystalline materials?
2. The observed WS step is slightly lower than the expected value. Is it possible that the step is higher than the theoretical value? Can the disorder of the sample be controlled here, and what is the influence of disorder on the experimental results?
3. For MAPbI₃, transient optical response is in fact dominated by the least dispersive direction of the band structure. Do other polycrystalline materials also have the same feature?
4. WS ladders and associated phenomena predicted and observed in biased semiconductor superlattices have intrigued scientists for decades (it seems the work of J. Bleuse et al., Phys. Rev. Lett. 60, 220, 1988, should be cited together with [4, 5]). The concept was later introduced and explored also in ultracold atoms (Phys. Rev. Lett. 76, 4512, 1996) and optical waveguide arrays superimposed with a linear optical potential (Opt. Lett. 23, 1701, 1998; Opt. Lett. 39, 1065, 2014). Since the paper is intended for an interdisciplinary journal like NC, the authors should discuss the broader impact of their work and possible connection to other fields.

Reviewer #3 (Remarks to the Author):

In this manuscript, the authors synthesized a MAPbI₃ polycrystalline film and studied ultrafast nonlinear optical responses of the MAPbI₃ film by performing THz-pump/NIR to visible-probe spectroscopy and theoretical calculations. They observed a transient absorption change induced by THz pump with a center frequency of 20 THz and assigned its origin to the Wannier-Stark localization. They found that the transient Wannier-Stark localization in a MAPbI₃ film can be observed at lower THz field strength than that in a single crystal GaAs, which results from a narrow electronic bandwidth and a large relevant lattice constant of MAPbI₃. The finding that its narrow electric bandwidth and large lattice constant make a MAPbI₃ film a unique optical nonlinear material would be important for developing novel photonic devices based on halide perovskites. However, the discussions are not enough to guarantee that the observed transient absorption change originates from the Wannier-Stark localization, on which I commented below. Therefore, my opinion is that this manuscript is not suitable for publication in Nature Communications as it stands.

The following points are the reasons why I doubt the interpretation of the data as the Wannier-Stark localization:

On page 4, the authors mentioned that the measured transient absorption change is well explained by a two-band model. However, I question why the higher energy states, such as light and heavy electron states, do not contribute to the optical nonlinear processes. In fact, the previous study [Z. Wei et al., Nat. Commun. 10, 5342 (2019).] reported that light and heavy electron states exist around 2.25 eV and the optical transitions between those states and the band-edge conduction band states significantly contribute to the two-photon absorption processes in a MAPbI₃ film. Therefore, I suspect that the observed transient absorption change originates from the Bloch-Siegert shift [E. J. Sie et al., Science 355, 1066 (2017).] in multiple states consisting of the band-edge valence and conduction band states and the higher energy conduction band states. Is it possible for the authors to comment on this?

How much is the spectral width of the THz pump pulses? As the authors stated on page 6, the phonon modes of halide perovskites fall in the low frequency range around several THz. Therefore, if the spectral width is larger than the phonon frequencies, impulsive stimulated Raman scattering

should occur. As some of the authors reported, driving the phonon modes leads to the similar transient absorption change albeit in a different crystal structure [Fig. 4a in H. Kim et al., Nat. Commun. 8, 687 (2017)]. Is it possible to discuss the contributions of phonons to the observed signals?

On page 13, the authors mentioned that the crystallites are arbitrarily oriented and the theoretical calculations were performed based on this assumption. Is there any possibility that some preferential orientations of the crystallites exist? Do the authors experimentally verify the assumption from, for example, XRD spectra? The detailed sample properties should be described in the Support Information, because the sample is not single crystal but complicated polycrystalline film.

On page 15, the authors claimed that the calculation result shown as a black curve in Fig. 4a is in good agreement with experimental results at high field amplitudes in Fig. 4b. However, the spectral oscillations around 2 eV can be seen only in the experimental results, not in the calculation. In addition, a bleaching signal around 2.2 eV which is expected from the calculation does not appear in the experimental results. What are the reasons for such disagreements?

I suggest some more points for improving manuscript:

On page 8, why did the authors consider the band gap energy of 2D perovskites with $l = 2$, not $l = 1, 3$ or 4 ?

With regard to Ref. 25, the authors should cite the published version [J.-C. Blancon et al., Nat. Commun. 9, 2254 (2018).], not that in arXiv.

Reviewer #1 (Remarks to the Author):

Berghoff et al report Wannier-Stark localization in a thin film of hybrid perovskite MAPbI₃ achieved via non-resonant excitation with a 20THz electric field. This is achieved in a self-assembled material and at low electric field intensities compared to previous results in GaAs. Given the potential applications in ultrafast optically controlled switching of material properties, the presented results are not only novel but also have a broad interest. The experiment involves ultrafast THz pump – optical probe spectroscopy and supported by extensive and well-crafted calculations, comparing the effect of static and transient electric fields and accounting for the effect of the sample polycrystallinity. This work represents an exciting contribution, and is suitable for publication in Nature Communications. A few minor points to be addressed are as follows:

1. It would be helpful to clearly note that in the data presented in the main manuscript (Fig.2(a)), the low-field regime refers only to the early delay times before the THz field peak. A separate measurement, with an overall lower THz field, could also be shown for comparison, providing a reference where no localization occurs.

→ First of all, we appreciate the reviewer for very encouraging and constructive comments. We agree with the reviewer's first suggestion and thus clarified the first regime as follows in the main text:

“More importantly, two distinct regimes can be identified in the time-resolved transient spectrum (Fig. 2(a)). The first regime appears at delay times $\tau < -100$ fs, where the field strength is relatively weak ($E < 3$ MV/cm), as an induced absorption (blue, $\Delta T/T < 0$) right below and an induced transmission (red, $\Delta T/T > 0$) right above the bandgap of $E_{\text{gap}} = 1.62$ eV. The second regime is apparent for field strengths $E > 3$ MV/cm, occurring between delay times $-100 < \tau < 100$ fs (Fig. 2(b)).”

Regarding the separate measurement, the low-field regime (< 2 MV/cm) has been analyzed in our previous work using a narrower-band optical probe [Nat. Commun. 8, 687] and in another reported work using \sim kHz field [ACS Photonics 2016, 3, 1060–1068]. There, one could find the electroabsorption spectra for comparison, where no localization occurs (Franz Keldysh effect).

Nonetheless, we have performed a separate measurement at a slightly lower peak field strength (4 MV/cm) with a different center frequency of 30 THz. In this case, we reproduce the observed transient Wannier Stark localization with lower peak field strength and different THz center frequency. We have added this result and the THz field characteristics in the supplementary figure S5.

2. The discussion of figure 2, which shows the main experimental finding of the paper and its interpretation, should be expanded. In particular, panel d is not sufficiently explained in the main text: how are the separate peaks appearing in the absorbance attributed to $n=0$, ± 1 ? The possibility of optically controlled switching between 3D materials and QWs is fascinating and could be explored more at length.

→ Yes, we agree that the detailed explanation of Figure 2 (c, d) would be necessary not only for clarifying the mechanism, but also for strengthening our interpretation. Therefore, we have added the following paragraph in the main text (experimental results):

“The extended structure of the transient spectral response can be understood with the assistance of Figure 2 (c, d). The localized Wannier-Stark states, equally spaced in energy by an amount $eE_{\text{THz}}D$, are depicted in the real-space along the field direction, z , in Fig 2 (c). D is the lattice period unit length and n the index. This space-dependent energy shift results in differentiating the electronic transition energies within the same site (arrow with $n = 0$, Fig 2 (c)) from those between different sites (arrows with $n = \pm 1$, Fig 2 (c)). As the difference in the transition energy with respect to the central spatially-direct ($n = 0$) transition is $neE_{\text{THz}}D$, one could assign the induced absorption below the band gap and above 1.9 eV to be $n=-1$ and $n=0$ transitions, respectively (Fig 2 (d)). The reduced absorption right above the band gap stems from the spectral transfer from non-perturbed optical transition to red- ($n=-1$) and blue- ($n=0$) shifted transitions (Fig 2 (d)). Depending on the strength of E_{THz} and the degree of localization, $|n| > 1$ transitions could, in principle, also be observed. In this case (Figure 2(a)), the observed single central step from reduced to increased absorption near the center of the band $E_{\text{pr}} = 1.9$ eV, is a noticeable signature of Stark localization, where the Wannier-Stark states are localized onto one unit cell.”

The possibility of optical switching between 3D delocalized and effective-2D localized electronic states is emphasized as follows:

“The large unit cell and the small bandwidths along one direction of this material allows for optical switching with up to ~ 40 % transmission modulation depth using relatively moderate biasing fields. Also, the optical modulation of the material is extremely fast (sub-20 fs), as demonstrated directly by the quasi-instantaneous response to an electric field oscillating at mid-infrared frequency.”

“The polycrystallinity of this material does not impede the optical switching performance of the material, since the least dispersive direction of the band structure dominates the contribution to the optical response, which favors low-cost fabrications. Together with the outstanding photophysical properties of MAPbI_3 , this finding highlights the potential of this material in novel photonic applications.”

3. Since the low-field onset of Wannier-Stark localization is a key claim of this work, is it possible to provide a confidence interval to the estimation of the THz electric field amplitude? Can it be absolutely calibrated e.g. using a thin (smaller than the coherence length between the EOS probe and all relevant THz wavelengths) electro-optic crystal?

→The reviewer raises an excellent point. In response, we have added the following statements to describe the accuracy of the THz electric field amplitude in the main text. What we would like to emphasize here is that the estimated field amplitude inside the material was even slightly over-estimated because we use the reported refractive index of the material at near-infrared frequency (2.2, by which the estimated peak field amplitude was 6 MV/cm as we report in the manuscript), which it is likely slightly higher in the measured frequency (20 THz). We note that the refractive index at 3 THz is reported to be ~ 3 (by which the estimated peak field amplitude is ~ 5 MV/cm). Therefore, we mention that we report the upper limit of the actual field strength.

“The phase-stable THz biasing fields are generated using a difference-frequency generation scheme in GaSe and characterized by ultrabroadband electro-optic sampling. The accuracy for determination of the absolute electric field strength in the center of the pump spot is estimated to be $\pm 15\%$. The peak field strength at the interior of the MAPbI_3 perovskite sample is obtained using the Fresnel transmission coefficient with the reported refractive index $n = 2.2$ at near-infrared frequency, which sets the upper limit of the actual field strength.”

The crystal we have used exhibits a rather flat response function in the relevant frequency range. Consequently, it does not distort the waveform significantly in the time domain. The crystal would be thinner than the coherence length for all relevant THz input frequencies. A detailed description of the procedure for determination of the field is found in the Methods section:

“The electric field transient is characterized by ultrabroadband electro-optic sampling at a 30- μm -thick GaSe crystal using balanced detection of an 8-fs probe pulse centered at a wavelength of 1.2 μm as the gating pulse. The quantitative value of the field amplitude is obtained by measuring the THz average power, pulse repetition rate and focal spot size. Then, the value at the interior of the MAPbI₃ perovskite sample is estimated using the Fresnel transmission coefficient for the THz field at the air–MAPbI₃ interface.”

4. Considering the in-depth discussion of orientations in both real space and Brillouin space in pages 12-13, a diagram not only of MAPbI structure in general but of the specific details being addressed would facilitate understanding (e.g. what Pb-I bond angles? Which direction is Γ M?)

→ We thank the reviewer for the helpful suggestion. Accordingly, we modified Fig 1a to indicate the exact Pb-I-Pb angle, the Pb-I bond length, and the most relevant direction (Γ M) mentioned in the main text on pages 12-13. Also, we refer to the figure when these details are mentioned (highlighted in the main text).

5. Is there a specific reason to consider MAPbI rather than other hybrid perovskites? Could one expect to easily replicate these results in other materials in the same family? Would a softer lattice, e.g. by selecting different cations or substituting a lighter metal for Pb, result in localization at even lower fields?

→ We appreciate the question highlighting the uniqueness of the MAPbI₃. In general, Wannier-Stark localization is very challenging to observe in natural crystals due to the small lattice constants, the rapid scattering, and the dielectric breakdown conditions. One of the most important messages of our work is that MAPbI₃, in particular, has the combination of the narrow bandwidth, the large periodicity, and the coincidence of the two directions. These unique features of its band structure promote this material to the Wannier-Stark regime under relatively modest field amplitudes. Therefore, if one could find any other material (including other perovskites) where these conditions are met, we could expect similar results, provided other material-specific disturbances such as interband tunnelings are absent. Regarding the softness of the lattice, if it means the lower nuclear vibration energy, there is not necessarily nor always a correlation between the electronic band dispersion near the band gap and a certain phonon frequency.

In the main text, we have tried to emphasize the unique electronic properties of MAPbI₃ that enable our observation, including the above-cited (comment #2) sentences:

“The large unit cell and the small bandwidths along one direction of this material allow for optical switching with up to ~ 40 % transmission modulation depth using relatively moderate biasing fields. Also, the optical modulation of the material is extremely fast (sub-20 fs), as demonstrated directly by the quasi-instantaneous response to an electric field oscillating at mid-infrared frequency.”

A few typos and mistakes in figure presentation should also be corrected:

* In the section “Experimental observation of Wannier-Stark Localization“, units are incorrect in the sentence: “relatively weak fields, $E < 3 \text{ MV}$, for $\tau < -100 \text{ fs}$ ”. It should read MV/cm .

*Is all the data presented (e.g. in fig. S2, S3) measured with a 6MV/cm THz field? Please specify so in captions.

* The x-axis of figure S1 is labeled incorrectly, it should probably be in eV and not nm

* Figures S4/S5 are not referred to in the main text

→ We thank the reviewer for thoroughly reviewing our materials and kindly raising the points that we overlooked. We corrected all of them accordingly (highlighted).

Reviewer #2 (Remarks to the Author):

This work provided an experimental observation of transient Wannier-Stark (WS) localization in a polycrystalline MAPbI₃ perovskite based on similar previous work performed in the GaAs/AlGaAs superlattice (Ref. 8 in this manuscript). As the authors emphasized, the main difference is that the field amplitude needed here (3MV/cm) is far less than that in GaAs/AlGaAs (which exceeds 10 MV/cm) due to the large relevant lattice constant, the small width of electronic energy bands, and the coincidence of these two along the same high-symmetry direction. The results and the method discussed here are interesting and the manuscript is well written. However, since similar work has been reported in literature and I did not find enough novelty of current work, I cannot accept the present paper for publication in high level journals like Nat Comm. The authors may want to show and discuss about the novelty and new highlights of their work should they try to submit to similar journals again. For instance:

1) Further analysis and emphasis of the peculiarity unique to MAPbI₃ (such as low field strengths required for WS), and in particular the applications of this feature.

→ First of all, we appreciate the reviewer's critical yet very helpful comments. For highlighting the first point, we have added the following paragraphs to the introduction, and have rewritten the conclusion and abstract:

[Introduction] "Besides their use in solar cells and light-emitting diodes, in this work, we demonstrate that MAPbI₃ also has a great potential in photonic applications, including optical modulators, optical switches, and optical signal processing. For any optical amplitude modulator, one of the essential properties is a substantial change of the absorption edge with relatively low required energies in general. We demonstrate that solution-processed, polycrystalline MAPbI₃ shows drastic changes in optical properties via Wannier Stark localization, at weak biasing fields. Whereas conventional semiconductors constituting photo-detectors, e.g. Si or InGaAs, require costly manufacturing processes and are limited to traditional rigid type devices, perovskites with distinct crystal structures exhibit ultrafast response (sub-20 fs), while simultaneously supporting cheap and flexible polycrystalline film fabrication.

...

The large unit cell and the small bandwidths along one direction of this material allows for optical switching with up to ~ 40 % transmission modulation depth using relatively moderate biasing fields. Also, the optical modulation of the material is extremely fast (sub-20 fs), as demonstrated directly by the quasi-instantaneous response to an electric field oscillating at mid-infrared frequency."

[Conclusion] "we have demonstrated that solution-processed, polycrystalline MAPbI₃ shows optical transmission change by tens of percent at relatively modest field strengths via transient Wannier Stark localization. The large lattice periodicity, the narrow electronic energy bandwidths, and the coincidence of these two along the same high-symmetry direction promotes this material to the Wannier Stark regime under relatively moderate biasing fields. Polycrystallinity of this material turns out not to hinder the Wannier Stark localization effect as observed, due to the dominant contribution from the least dispersive direction of the band structure, which favors low-cost fabrications with this material as optical modulators. The degree of disorder and relative orientation among crystallites may influence the modulation spectral shape slightly, e.g. the position of the photon energy where the induced transmission to induced absorption happens, which could be finely tuned depending on the desired device performance by further systematic studies."

[Abstract] “Methylammonium lead iodide perovskite (MAPbI_3), renowned for an impressive power conversion efficiency rise and cost-effective fabrication for photovoltaics, exhibits a huge potential for optical modulation-type applications, in this work. We demonstrate that polycrystalline MAPbI_3 s undergo drastic changes in optical properties with the modulation depth to be tens of percent at moderate field strengths, via transient Wannier Stark localization with an ultrafast response time. The distinct band structure of this material - the large lattice periodicity, the narrow electronic energy bandwidths, and the coincidence of these two along the same high-symmetry direction – enables relatively weak fields to bring this material into the Wannier Stark regime. Its polycrystalline nature is not detrimental to the optical switching performance of the material, since the least dispersive direction of the band structure dominates the contribution to the optical response, which favors low-cost fabrication. Together with the outstanding photophysical properties of MAPbI_3 , this finding highlights the potential of this material in novel photonic applications.”

2) Any other typical physical phenomena not limited to WS localization that are based on the transient spectral analysis? The authors developed a direct method to show WS localization in real natural solids. Since such localization has already been extensively studied in previous work, it will be interesting if this method can also be used for the study of other physical phenomena.

→ We thank the reviewer for the suggestion to emphasize the advantage and potential use of our experimental technique and approach. Accordingly, we have added a dedicated paragraph in the conclusion as follows:

“Moreover, the phase stable THz field transients and the ultra-broadband optical pulses of 7 fs duration revealed that the optical modulation of this material has an extremely fast, quasi-instantaneous (sub-20fs) temporal response in visible/near-IR spectral region. This technique could be generalized for realizing transient Wannier Stark localization in other semiconductor solids in a carefully prepared single-crystalline or a polycrystalline form. More generally, this method enables to analyze any ultrafast changes in optical properties induced by the phase-locked and intense electromagnetic field transients, be it resonantly or non-resonantly. While here we used only the electric field of the transients, one could also exploit the magnetic component for exploring ultrafast magneto-optic effects, by enhancing the magnetic field with respect to the electric field with, e.g., a specially designed plasmonic nanoaperture.”

3) MAPbI_3 perovskite exhibits wide light absorption range and excellent photo-electronic properties and is considered as a prominent light harvester. In this manuscript, the description and discussion of novel properties of MAPbI_3 , especially related to this work, seems absent.

→ We thank the reviewer for this suggestion for improving our work. We have added the following paragraph to the introduction to better connect to the cited part of the conclusion:

[Introduction] “Methylammonium lead iodide perovskite (MAPbI_3) has become a remarkable material for photovoltaic applications due to the dramatic increase of the power conversion efficiency and the cost-effective fabrication processes. The success of this material has been attributed to large absorption coefficient and the exceptional transport properties such as long carrier diffusion length, high carrier mobilities and defect tolerance.”

[Conclusion] *“Together with the renowned photophysical properties of MAPbI₃, such as the long carrier diffusion length, low mid-gap trap density, and large absorption coefficient, this finding of high modulation depth, ultrafast response, and low onset field for Wannier-Stark localization highlights the potential of this material in photonic applications.”*

Other questions and suggestions:

1. In this work, the WS localization is created by applying the bias field from the phase-stable THz pulse and can be detected by optical absorption spectra. Is this method generally applicable for single-crystalline and polycrystalline materials?

→ Yes, it could be generally applicable for single-crystalline and polycrystalline materials. Thus, we added this sentence in the conclusion:

“This technique could be generalized for realizing transient Wannier Stark localization in other semiconductor solids in a carefully prepared single-crystalline or a polycrystalline form.”

2. The observed WS step is slightly lower than the expected value. Is it possible that the step is higher than the theoretical value? Can the disorder of the sample be controlled here, and what is the influence of disorder on the experimental results?

→ First, we would like to point out that the predicted position where the WS step occurs depends on the level of theory. If we compare the Fig 3 and Fig 4(c, d), the step occurs at slightly below 2 and 1.9 eV, respectively, and the experimentally observed position was 1.9 eV. The difference in the theoretical approaches to obtain Fig 3 and Fig 4(c,d) is whether the other direction is taken into account (i.e., with (Fig 4(c,d)) and without (Fig 3) the averaging over the disorder). The disorder of the sample could not be controlled experimentally here, but from this theoretical comparison, what one could learn about the influence of disorder on the experimental results would be indeed the position of the WS step (with ~100 meV range). We strengthened this discussion by adding the following to the discussion and conclusion sections:

[Discussion] *“This finding is remarkable because it means the two extreme cases – the completely random orientation of a polycrystalline sample and the perfectly oriented single crystal – are expected to produce very similar optical responses. The only slight difference between the two extremes would be a small shift in the photon energy (~100 meV) where the induced transmission turns to the induced absorption and in the transient spectral shape. The two extremes include partial preferential orientations. We also note that the averaging process using only two extreme directions does not contain any material-specific information, which means that one could expect other polycrystalline materials to behave similarly.”*

[Conclusion] *“The degree of disorder and relative orientation among crystallites may influence the modulation spectral shape slightly, e.g. the position of the photon energy where the induced transmission to induced absorption happens, which might be finely tuned depending on the desired device performance by further systematic studies.”*

3. For MAPbI₃, transient optical response is in fact dominated by the least dispersive direction of the band structure. Do other polycrystalline materials also have the same feature?

→ Yes, we believe so, because for the larger bandwidth, the field required for WS localization will be larger. This means that the direction with the smallest dispersion will always be where WS

localizations occur for the smallest field. While we observe the localization along the least dispersive direction, the contributions from the other directions are negligible, since the transmission changes are smaller for fields below the WS threshold. It is indeed what we observe from the theoretical model, including averaging over the arbitrarily oriented crystallites. Regarding this point, we added this sentence in the main text (discussion section):

“We also note that the averaging process using only two extreme directions does not contain any material-specific information, which means that one could expect other polycrystalline materials to behave similarly.”

4. WS ladders and associated phenomena predicted and observed in biased semiconductor superlattices have intrigued scientists for decades (it seems the work of J. Bleuse et al , Phys. Rev. Lett. 60, 220, 1988, should be cited together with [4, 5]). The concept was later introduced and explored also in ultracold atoms (Phys. Rev. Lett. 76, 4512, 1996) and optical waveguide arrays superimposed with a linear optical potential (Opt. Lett. 23, 1701, 1998; Opt. Lett. 39, 1065, 2014). Since the paper is intended for an interdisciplinary journal like NC, the authors should discuss the broader impact of their work and possible connection to other fields.

→ We thank the reviewer for this very insightful suggestion to position our work in a broader context and for the listed literature. We have, in our revised manuscript, cited and discussed the suggested literature and make a connection in the introduction as follows:

“Following the initial observations in semiconductor superlattices under static bias fields⁸⁻¹⁰ Wannier-Stark ladders have been proposed and realized in various physical systems featuring wave propagation in the presence of periodic potentials and a homogeneous force. Examples include ultracold atoms in an accelerating 1D standing wave⁵, waveguide arrays with linearly varying propagation constants⁶, and self-accelerating optical beams in 1D photonic lattice⁷. Several fundamental observations and device applications from the Wannier-Stark localization have been focused on statically biased artificial semiconductor superlattices⁸⁻¹². However, in natural homogeneous solids, where the periodicity is dictated by the atomic structure, such an extreme state of matter has never been achieved using static biasing. To resolve optical transitions to individual Wannier-Stark states in, e.g., absorption spectra, their energetic spacing needs to be larger than the (total) linewidth Γ , i.e., $eED > \Gamma$ ^{9,10,13} Due to the small lattice constant of bulk crystals and the large linewidth which results from the scattering of electrons with lattice vibrations and other electrons, the requirement $eED > \Gamma$ can typically not be fulfilled under stationary external fields below the strength where the dielectric breakdown occurs^{11,12}.”

Reviewer #3 (Remarks to the Author):

In this manuscript, the authors synthesized a MAPbI₃ polycrystalline film and studied ultrafast nonlinear optical responses of the MAPbI₃ film by performing THz-pump/NIR to visible-probe spectroscopy and theoretical calculations. They observed a transient absorption change induced by THz pump with a center frequency of 20 THz and assigned its origin to the Wannier-Stark localization. They found that the transient Wannier-Stark localization in a MAPbI₃ film can be observed at lower THz field strength than that in a single crystal GaAs, which results from a narrow electronic bandwidth and a large relevant lattice constant of MAPbI₃. The finding that its narrow electric bandwidth and large lattice constant make a MAPbI₃ film a unique optical nonlinear material would be important for developing novel photonic devices based on halide perovskites. However, the discussions are not enough to guarantee that the observed transient absorption change originates from the Wannier-Stark localization, on which I commented below. Therefore, my opinion is that this manuscript is not suitable for publication in Nature Communications as it stands.

The following points are the reasons why I doubt the interpretation of the data as the Wannier-Stark localization:

On page 4, the authors mentioned that the measured transient absorption change is well explained by a two-band model. However, I question why the higher energy states, such as light and heavy electron states, do not contribute to the optical nonlinear processes. In fact, the previous study [Z. Wei et al., Nat. Commun. 10, 5342 (2019).] reported that light and heavy electron states exist around 2.25 eV and the optical transitions between those states and the band-edge conduction band states significantly contribute to the two-photon absorption processes in a MAPbI₃ film. Therefore, I suspect that the observed transient absorption change originates from the Bloch-Siegert shift [E. J. Sie et al., Science 355, 1066 (2017).] in multiple states consisting of the band-edge valence and conduction band states and the higher energy conduction band states. Is it possible for the authors to comment on this?

→ First of all, we appreciate the reviewer's critical, yet very constructive comments. We also thank the reviewer for raising this possible alternative explanation, based on which we were able to improve our manuscript substantially. Here we clarify the detailed assignment of each spectral component (in our optical response) based on our mechanism (WS localization), which seem inconsistent with other nonlinear optical processes, and strengthen our interpretation as follows:

“The extended structure of the transient spectral response can be understood with the assistance of Figure 2 (c, d). The localized Wannier-Stark states, equally spaced in energy by an amount $eE_{\text{THz}}D$, are depicted in the real-space along the field direction, z , in Fig 2 (c). D is the lattice period unit length and n the index. This space-dependent energy shift results in differentiating the electronic transition energies within the same site (arrow with $n = 0$, Fig 2 (c)) from between different sites (arrows with $n = \pm 1$, Fig 2 (c)). As the difference in the transition energy with respect to the central spatially-direct ($n = 0$) transition is $neE_{\text{THz}}D$, one could assign the induced absorption below the band gap and above 1.9 eV to be $n = -1$ and $n = 0$ transitions, respectively (Fig 2 (d)). The reduced absorption right above the band gap stems from the spectral transfer from non-perturbed optical transition to red- ($n = -1$) and blue- ($n = 0$) shifted transitions (Fig 2 (d)). Depending on the strength of E_{THz} and the degree of localization, $|n| > 1$ transitions could, in principle, also be observed. In this case (Figure 2(a)), the observed single central step from reduced to increased

absorption near the center of the band $E_{pr} = 1.9$ eV, is a noticeable signature of Stark localization, where the Wannier-Stark states are localized onto one unit cell.

This observed transient response is clearly distinct from the shift of transition energy due to the presence of photon-dressed states²⁸. Photon-dressed states can shift energies in only one direction (mostly blue-shift) with energy orders of magnitude smaller than the bandwidth. Also, the contribution from the possible higher energy bands within our probe photon energy range²⁹ is negligible. Indeed, we neither observe any additional Franz-Keldysh and/or Wannier-Stark response within our probe energy, nor find any decay of the entire signal as a function of the field strength due to the tunneling to higher energy bands at intense field regime. Therefore, we could consider simple two-band systems to understand our experimental demonstration of Wannier-Stark localization in further detail. As will be shown below, the two-band model explains our observations.”

Regarding the first point (in slightly more detail), given the presence of the higher band with the energy difference of around 2.25 eV, there are two possibilities of experimental features one could observe in our result.

The first possibility is the case of tunneling to a higher energy band in the presence of strong THz fields. In that case, the photoexcited electrons and holes could tunnel into an energetically higher band and would then not contribute to WS localization in the lowest bands. Also, the signal could possibly decay as a function of the field amplitude for strong fields (since the tunneling rate has an exponential dependence on the field amplitude). However, we do not see indications in the experiment that this is happening to a significant degree.

In the other possibility, where the tunneling does not occur, if WS localization is also not the origin of our experimental observation, then each band would have shown at least Franz Keldysh effect at one around the primary bandgap of 1.6 eV and the other one at around the secondary gap above 2.25 eV (as the TPA paper reports). Since the Franz Keldysh effect appears as an oscillatory feature only near the gap (within 100 meV in probe photon energy), in that case one would expect two separate Franz Keldysh features - one in the 1.5~1.7 eV range and another at 2.1~2.3 eV, with no response for energies in between (1.7~2.1 eV). In contrast, we find an induced transparency from 1.6 eV all the way up to 1.9 eV without interruption. The reason we did not observe any response in 2.2eV (even in FKE) could be understood from the different transition probability in the one-photon and two-photon transition processes. Our optical probing scheme involves one-photon transition where the response around 2.2eV is much weaker than the case of two-photon transition. Therefore, we could safely rule out the other scenario, too.

Regarding the second point, the Bloch-Siegert shift is typically very small in magnitude < 1 eV and, in exceptional cases at best comparable to the optical Stark shift (~ 10 meV in the mentioned literature). Also, such a shift happens in only one direction: typically only blue shift (because this effect originates from the state repulsion). However, in our case, we have induced absorption below and above the band gap, which means that both red-shifted and blue-shifted transitions occur simultaneously, and reduced absorption at around the band gap due to the spectral transfer. These features are unique to Wannier Stark localization.

How much is the spectral width of the THz pump pulses? As the authors stated on page 6, the phonon modes of halide perovskites fall in the low frequency range around several THz. Therefore, if the spectral width is larger than the phonon frequencies, impulsive stimulated Raman scattering should occur. As some of the authors reported, driving the phonon modes leads to the similar transient absorption change albeit in a different crystal structure [Fig. 4a in H. Kim et al., Nat. Commun. 8, 687 (2017).]. Is it possible to discuss the contributions of phonons to the observed signals?

→ The spectral width of the THz pump pulses is ~ 4 THz. We added the spectrum of the THz pulses of the main result (fig 2a) in the supplementary information (fig. S1). For comparison, we also added the Fourier transform of the time profile in fig 2a in the SI (fig. S4).

As evident in this spectrum (fig S4), the main oscillation around time zero is 40 THz, and there is no distinct oscillation below 4 THz except for the slow envelope of the pulse. More importantly, if there is any phonon coherence, we know that the dephasing times of at least 1 and 2 THz modes are 0.3 and 1 ps, respectively, and both are much longer than the duration of the IR pulse and our experimental window. So if that was the case, we should have seen responses after the overlap of IR pulses and the visible probe (after $t = 200$ fs). However, we do not see any oscillation immediately after that, which indicates no contribution from phonons within the bandwidth. Thus, the impulsive stimulated Raman scattering could be safely ruled out. We added this argument on the main text as follows:

“It is noteworthy that the bandwidth of the THz pulse is ~ 4 THz (< 40 THz modulation, Fig. S4), so that in principle impulsive stimulated Raman excitation of sub-4 THz modes be possible. However, no oscillatory signal was observed after 150 fs, which is much shorter than the dephasing times of reported phonon modes with frequency up to 4 THz. Therefore, any possibility of coherent phonon contribution to the temporal modulation can be ruled out.”

On page 13, the authors mentioned that the crystallites are arbitrarily oriented and the theoretical calculations were performed based on this assumption. Is there any possibility that some preferential orientations of the crystallites exist? Do the authors experimentally verify the assumption from, for example, XRD spectra? The detailed sample properties should be described in the Support Information, because the sample is not single crystal but complicated polycrystalline film.

→ We understand and appreciate the reviewer’s concern. However, any possibility of preferential orientations of the crystallites does not make any critical effect, as the two extreme cases – completely arbitrary orientation and perfectly single orientation – produce more or less the same THz induced optical spectra. From this theoretical approach, what we learn is that, no matter how much preferential orientation we have (from zero to complete), the contribution from the least dispersive direction dominates the optical response. Although we indeed can not completely rule out the possibility of partial preferential orientation, this possibility does not affect our conclusion. We highlight the point by adding these sentences in the main text:

“The results of Fig. 4(a, b) suggest that, for the randomly oriented crystallites in the film, the overall response is dominated by the response originating from the band dispersion in the $\bar{1}\bar{1}\bar{2}$ direction. This reasoning is substantiated by the averaged field-dependent absorption changes calculated for both a static and a THz field shown in Figs. 4(c) and (d), respectively. This finding is remarkable because it means the two extreme cases – the completely random orientation of a polycrystalline

sample and the perfectly oriented single crystal – are expected to produce very similar optical responses. The only slight difference between the two extremes would be a small shift in the photon energy (~100 meV) where the induced transmission turns to the induced absorption and in the transient spectral shape. The two extremes include partial preferential orientations.”

Regarding our sample, we added the SEM image of the polycrystalline perovskite MAPbI₃ film spin-coated on TOPAS® substrate in the supplementary information (fig S2). We did not measure the XRD data of the measured sample, but we do have the XRD spectra of the perovskite polycrystalline film prepared in the exactly same way [fig S15 of J. Phys. Chem. Lett. 2015, 6, 4991]. There, one could find not only (hh0) but also other Bragg peaks. Besides, even in the case of a complete preferential orientation of a surface, still the most critical orientation is the relative angle between the [001] direction of each crystallite and the THz field polarization, which is not possible to define. The orientations of [001] direction of each crystallite on top of a completely disordered substrate can still be assumed to be arbitrarily oriented and difficult to measure with the currently available resolution of X-ray microscopy (~500 nm in J. Phys. Chem. C. 2017, 121, 7596).

On page 15, the authors claimed that the calculation result shown as a black curve in Fig. 4a is in good agreement with experimental results at high field amplitudes in Fig. 4b. However, the spectral oscillations around 2 eV can be seen only in the experimental results, not in the calculation. In addition, a bleaching signal around 2.2 eV which is expected from the calculation does not appear in the experimental results. What are the reasons for such disagreements?

→ We thank the reviewer for thoroughly examining our results. Yes, the first point, the spectral oscillations around 2 eV, is indeed something we have tried to understand. The oscillation (or repeated peaks) has an interval of 80~100 meV regardless of the THz field strength. Interestingly this interval matches well with our THz photon energy (83 meV). The oscillations are quite stable as function of the field amplitude, however, they do not correspond to Franz-Keldysh and neither to WS. To analyze the possible origin of these oscillations, we did perform additional model calculations. To that end, we, in particular, modified the k/energy dependence of the interband dipole and also considered a complex interband dipole with a symmetric real and an anti-symmetric imaginary part (it was recently shown that such an interband dipole with such a k-dependence gives rise to SHG in two-band models). With such phenomenological model extensions, we do obtain signatures having similarities with THz sidebands in the optical spectra, see Fig. S9 in the supplementary information. However, the obtained results are quite different from the oscillations observed in the experiment and the model assumptions required to obtain them are rather unrealistic.

Regarding the second point, in a 1D model with WS localization, we remove absorption from the band edges, and this concentrates in the band center. Since we average over systems of different bandwidths and lattice constants, the critical field where the transition to WS takes places and also the position of the band center is not fixed but depends on the individual case, i.e., differs for the different f's. As a result, in Fig. 4(c) and 4(d) the transitions from blue- to red- shift appear at higher energies with increasing field amplitude, since with increasing amplitude also systems with larger bandwidth, i.e., larger f, enter the WS regime. Therefore, with any two-band model one should always see some bleaching near the upper band edge due to the concentration, or in other words, a shift of the oscillator strength towards the center of the band.

In any case, the precision of our simple modeling involving some fitting of DFT results with few parameters gets more inaccurate with increasing photon energies.

To clarify this, we add an explanation in the main text as follows:

“The averaged graph is in good agreement with the differential spectra at high field amplitudes given very few parameters to describe the entire contributions from complicated actual band structure (i.e. two extreme bandwidths, lattice constants, and linear interpolation of them). The prediction of 1.9 eV, almost exactly where the change from bleaching to induced absorption is observed in the experiment (Fig 4(b), upper curves) is remarkable. A few minor features including the shape at higher photon energy could be improved by modifying the model band curve and oscillations around 2 eV in Fig 4(b) need further studies (Fig S9). More importantly, this position of 1.9eV is very close to the center of the band structure for the $\overline{\Gamma Z}$ direction, 2 eV as the single direction model indicated above.”

I suggest some more points for improving manuscript:

On page 8, why did the authors consider the band gap energy of 2D perovskites with $l=2$, not $l=1, 3$ or 4 ?

→ It is because of the doubling of the unit cell in the tetragonal phase compared to the cubic phase. It means the double layers ($l=2$) Pb-I octahedra is the repeating unit, and in turn the spatial range of confinement. That was the reason we particularly compared the probe energy where one could observe the abrupt change of transmittance and the band gap energy of 2D perovskite with $l=2$. We realized it could be confusing without further clarification. As such, we modified the corresponding sentence in the discussion section as following:

“In the case of $(BA)_2(MA)_{l-1}PbI_{3l+1}$ perovskites, where the PbI_6 octahedral network forms a double layer ($l = 2$), the same periodicity of the sample along the c axis, the optical band gap is ~ 2.1 eV, which is comparable to the observed 1.9 eV.”

With regard to Ref. 25, the authors should cite the published version [J.-C. Blancon et al., Nat. Commun. 9, 2254 (2018).], not that in arXiv.

→ We cited this reference accordingly (highlighted).

Reviewers' comments:

Reviewer #1 (Remarks to the Author):

The authors have addressed all my concerns/comments in the revised manuscript. However, there are two important concerns from other reviewers which are very important. One is the role of the polycrystallinity of the sample. The only way to address this is to perform measurements on single crystals or two have extensive structural characterization of the sample, which are out of the scope of current work. Publication of this manuscript will inspire future work in that direction, I reckon.

The second concern, as reviewer 3 rightly points out, is with regards to possible alternative explanations to the presented experimental results. The authors provide a convincing arguments to discount other possibilities. Some of the explanations provided in the response to reviewer 3 were not included in the manuscript, for example, on why Bloch-Siegert is not a viable explanation. It may be very instructional to expand on this discussion in the main manuscript.

Apart from these very minor comments, the manuscript is suitable for publication in Nature Communications.

Reviewer #2 (Remarks to the Author):

I have read through authors' response, and the revised manuscript. I'm okay with authors' response to my comments, and thus I don't have further objection to accept this manuscript for publication in NC.

Reviewer #3 (Remarks to the Author):

I confirmed the replies from the authors, the revised manuscript, and the supporting information. I appreciate the efforts of the authors to answer my questions and comments. However, I still do not think that it is convincing that the origin of the observed nonlinear responses can be attributed to the Wannier-Stark localization. As the authors replied, the theoretical curves shown in Fig. 4a do not fully reproduce the experimental results in Fig. 4b partly because the theoretical model is too simple. Therefore, to make the interpretation as the Wannier-Stark localization solid, it is important to rule out the other possible origins. Although the other possibilities are discussed in the reply letter and the revised manuscript, I am not satisfied with some of those discussions. Thus, I do not recommend publication of the manuscript for Nature Communications as it stands. I suggest that the following points should be considered.

1. On page 9 in the main manuscript, the authors mentioned that the two induced absorption signals shown in Fig. 2d can be assign to $n = -1$ and $n = 0$ transitions, whose difference in the transition energy is $eE _{THz}D$. How much is the value of $eE _{THz}D$? In addition, I cannot read from Fig. 2d what the $n = -1$ and $n = 0$ transition energies are.

2. On page 9 in the main manuscript, the authors stated "Photon-dressed states can shift energies in only one direction (mostly blue-shift) ..." However, I do not agree with the statement. If the higher energy levels exist, the Autler-Townes effect can shift the band-edge transitions in both directions, i.e. blue- and red- shifts, as reported in [C.-K. Yong et al. Nat. Mater. 18, 1065–1070 (2019).]. In addition, [G. Yumoto et al. Nat. Commun. 12, 3026 (2021).] recently reported that three-level Autler-Townes effect indeed occurs in halide perovskites. How do the authors comment on the possibility that the Autler-Townes effect can be the origin of the observed optical nonlinear responses?

3. On page 10 in the main manuscript, the authors stated "..., nor find any decay of the entire signal as a function of the field strength due to the tunneling to higher energy bands at intense field regime." I agree to the authors that the real excitation of carriers cannot explain the data due to the absence of the signal decay. However, the virtual population generated only under the THz-field irradiation should induce additional transitions for probe pulses, which would result in induced absorption signals. Is it possible to comment on this?

Reviewer #1 (Remarks to the Author):

The authors have addressed all my concerns/comments in the revised manuscript. However, there are two important concerns from other reviewers which are very important. One is the role of the polycrystallinity of the sample. The only way to address this is to perform measurements on single crystals or two have extensive structural characterization of the sample, which are out of the scope of current work. Publication of this manuscript will inspire future work in that direction, I reckon.

The second concern, as reviewer 3 rightly points out, is with regards to possible alternative explanations to the presented experimental results. The authors provide a convincing arguments to discount other possibilities. Some of the explanations provided in the response to reviewer 3 were not included in the manuscript, for example, on why Bloch-Siegert is not a viable explanation. It may be very instructional to expand on this discussion in the main manuscript.

Apart from these very minor comments, the manuscript is suitable for publication in Nature Communications.

→ First of all, we appreciate the reviewer for very encouraging and constructive comments. We agree with the reviewer's suggestion, and thus, we explicitly include a discussion explaining why these scenarios can be ruled out in this revised manuscript as follows:

“It is important to distinguish this transient Wannier Stark localization from the optical Stark-type effects such as the Autler-Townes effect and the Bloch-Siegert shift. In general, an external electric field affects the optical properties of a semiconductor in two ways: there are spectral and kinetic aspects. Spectral aspects refer to energy shifts and broadenings that arise from mixing two states by the external optical field. The mixing of the wavefunctions results in dressed states and leads to the Stark-type shifts. The magnitude of such shifts increases with the amplitude of the incident field and the interband dipole matrix, but decreases with increasing detuning between the light frequency and the transition frequency. On the other hand, kinetic aspects represent the evolution of the particle distributions in the renormalized states driven by the external field, which is called “intraband acceleration”. This intraband acceleration leads to the Franz Keldysh effect at a moderate field strength and eventually Wannier-Stark localization in the strong-field regime.

Each of these two contributions can be straightforwardly separated in the semiconductor Bloch equations (SBE). Specifically, the third and second terms on the right-hand side of the Eq (1) represent the spectral (optical Stark-type effects) and the kinetic (Wannier Stark localization) aspects, respectively. One can thus directly compare each contribution to the differential optical response. As evident from Supplementary Fig. 6, the THz-induced optical Stark effect is shown to be much weaker (on the order of a few meV) compared to the shifts arising from the Wannier Stark localization which corresponds to approximately half the band width (several 100 meV). Therefore, we conclude that the observed transient response is mainly contributed from the Wannier Stark localization.”

Reviewer #2 (Remarks to the Author):

I have read through authors' response, and the revised manuscript. I'm okay with authors' response to my comments, and thus I don't have further objection to accept this manuscript for publication in NC.

→ We thank the reviewer very much for the positive comments.

Reviewer #3 (Remarks to the Author):

I confirmed the replies from the authors, the revised manuscript, and the supporting information. I appreciate the efforts of the authors to answer my questions and comments. However, I still do not think that it is convincing that the origin of the observed nonlinear responses can be attributed to the Wannier-Stark localization. As the authors replied, the theoretical curves shown in Fig. 4a do not fully reproduce the experimental results in Fig. 4b partly because the theoretical model is too simple. Therefore, to make the interpretation as the Wannier-Stark localization solid, it is important to rule out the other possible origins. Although the other possibilities are discussed in the reply letter and the revised manuscript, I am not satisfied with some of those discussions. Thus, I do not recommend publication of the manuscript for Nature Communications as it stands. I suggest that the following points should be considered.

→ We appreciate the reviewer for the thorough review and for giving us the opportunity to improve our manuscript even further. We apologize for apparently underestimating this point: we mistakenly felt it inappropriate to devote a substantial part of the discussion in the manuscript to this point. Rather, we tried to convince the reviewer in the rebuttal. However, we take this concern more carefully into account this time, and explicitly include a discussion explaining what are the similarities and differences between these optical Stark-type effects (i.e. Autler-Townes effects and Bloch-Siegert effects) in a theoretical context and how exactly each effect contributes to the experimentally observed optical response. This was also suggested by reviewer #1.

In the strong field regime (like our experiment), the Wannier-Stark localization (WSL) shifts the onset of the energy gap by approximately half the bandwidth, i.e., several hundred meV. Since the THz frequency is much lower than the interband transition frequencies, the optical Stark shift can be described perturbatively and is determined by the square of the Rabi frequency divided by the detuning. For a maximal field amplitude of 10 MV/cm and the interband dipole matrix element, the resulting Stark shifts are on the order of just a few meV, which is orders of magnitude smaller than the shifts arising from the WSL. These arguments are in agreement with and supported by new simulations in which we artificially neglected the intraband acceleration in order to study Stark-type shifts separately (see Supplementary Figure 6 and Supplementary Method).

Therefore, we could further confirm that the Wannier-Stark localization (originating from the intraband acceleration driven by the field) dominates the observed differential transmission. In contrast, the optical Stark-type effects (which leads to the spectral shift of each k-state of the band) have a negligible amplitude compared to the WSL.

We thank the reviewer for the opportunity to revisit the fundamentals and strengthen our interpretation. Further details are covered in our reply to comment #2 below.

1. On page 9 in the main manuscript, the authors mentioned that the two induced absorption signals shown in Fig. 2d can be assigned to $n = -1$ and $n = 0$ transitions, whose difference in the transition energy is $eE_{\text{THz}}D$. How much is the value of $eE_{\text{THz}}D$? In addition, I cannot read from Fig. 2d what the $n = -1$ and $n = 0$ transition energies are.

→ We agree with the reviewer that the value of energy distance $eE_{\text{THz}}D$ is only inferred, and the necessary numbers appear in a rather scattered way (e.g. $E_{\text{THz}} = 6$ MV/cm in page 11, and $D =$ the largest lattice constant along the c axis, 12.5 \AA , in page 15). Therefore, we added the figure caption of figure 2d the following sentence:

“In case of $E_{\text{THz}} = 6$ MV/cm in considering the lattice constant D of 12.5 \AA , $\Delta E_{\text{WSL}} = eE_{\text{THz}}D$ is estimated to be 750 meV , consistent with the spectrum showing that the absorption band of $n = -1$ and $n = 0$ are approximately 750 meV apart.”

Indeed, the $n=-1$ and $n=0$ transitions have certain energy ranges rather than a single transition energy (i.e. two-level systems) since these are “interband” transitions among mini-bands. So the range of transition energies for $n=-1$ is $1.2\sim 1.62\text{eV}$, and that for $n=0$ is $1.95\sim 2.38 \text{ eV}$, and these transition energy ranges are approximately 750 meV apart (figure 2d).

2. On page 9 in the main manuscript, the authors stated “Photon-dressed states can shift energies in only one direction (mostly blue-shift) ...” However, I do not agree with the statement. If the higher energy levels exist, the Autler-Townes effect can shift the band-edge transitions in both directions, i.e. blue- and red- shifts, as reported in [C.-K. Yong et al. Nat. Mater. 18, 1065–1070 (2019).]. In addition, [G. Yumoto et al. Nat. Commun. 12, 3026 (2021).] recently reported that three-level Autler-Townes effect indeed occurs in halide perovskites. How do the authors comment on the possibility that the Autler-Townes effect can be the origin of the observed optical nonlinear responses?

→ We agree with the reviewer that the optical Stark effect (i.e. the Autler-Townes effect) in fact leads to energy splitting. This initially two-level system with energies of ϵ_1 and ϵ_2 undergoes light-field induced level splitting to $\epsilon_1 + \frac{\nu}{2} \pm \frac{1}{2}\sqrt{\nu^2 + \hbar^2\omega_R^2}$ and $\epsilon_2 - \frac{\nu}{2} \pm \frac{1}{2}\sqrt{\nu^2 + \hbar^2\omega_R^2}$ respectively, where ν is detuning ($\epsilon_2 - \epsilon_1 - \hbar\omega_{\text{laser}}$) and ω_R is the Rabi frequency. The transitions among these split states are observed from singlet (without optical field) to triplet (“Mollow triplet”, with optical field). In case of weak excitation and finite detuning (here we make a Taylor expansion of the square roots, so it’s a perturbative result), the upper sideband $\epsilon_2 - \epsilon_1 + \frac{1}{2}\frac{\omega_R^2}{\nu}$ is closest to the original resonance. This transition is the one that is observed in the experiment as a small blueshift depending on the intensity of the light field (and the interband dipole matrix element), if the frequency of the exciting light is smaller than the transition frequency. Although by changing the detuning (frequency difference between the light frequency and the transition frequency) it is possible to observe the peak splitting as the reviewer has pointed out, the blue shift is the common signature for large detuning (such as our experiment - for THz field the detuning to an optical transition is very large). [H. Haug & S. W. Koch, “Quantum Theory of the Optical and Electronic Properties of Semiconductors”]

However, more fundamentally, in a semiconductor, unlike two-level systems (or multi-levels without k-space dispersion), an external field has pronounced influence on the relative motion of optically generated electron-hole pair, going well beyond the field-induced shifts of excitonic resonance. And this is what we demonstrate by separating the contribution of the optical Stark effect and the intraband acceleration: WSL is by far the dominating contribution to the observed differential optical response and produces the spectral shape in very good agreement. In contrast, the optical Stark-type contributions (Autler-Townes effect from the corotating field and Bloch Siegert shift from the counterrotating field) are almost negligible and, moreover, do not agree with the observed spectral shape. Therefore, we added the following paragraph and the supplementary figure 6 showing this additional result:

“It is important to distinguish this transient Wannier Stark localization from the optical Stark-type effects such as the Autler-Townes effect and the Bloch-Siegert shift. In general, an external electric field affects the optical properties of a semiconductor in two ways: there are spectral and kinetic aspects. Spectral aspects refer to energy shifts and broadenings that arise from mixing two states by the external optical field. The mixing of the wavefunctions results in dressed states and leads to the Stark-type shifts. The magnitude of such shifts increases with the amplitude of the incident field and the interband dipole matrix, but decreases with increasing detuning between the light frequency and the transition frequency. On the other hand, kinetic aspects represent the evolution of the particle distributions in the renormalized states driven by the external field, which is called “intraband acceleration”. This intraband acceleration leads to the Franz Keldysh effect at a moderate field strength and eventually Wannier-Stark localization in the strong-field regime.

Each of these two contributions can be straightforwardly separated in the semiconductor Bloch equations (SBE). Specifically, the third and second terms on the right-hand side of the Eq (1) represent the spectral (optical Stark-type effects) and the kinetic (Wannier Stark localization) aspects, respectively. One can thus directly compare each contribution to the differential optical response. As evident from Supplementary Fig. 6, the THz-induced optical Stark effect is shown to be much weaker (on the order of a few meV) compared to the shifts arising from the Wannier Stark localization which corresponds to approximately half the band width (several 100 meV). Therefore, we conclude that the observed transient response is mainly contributed from the Wannier Stark localization.”

3. On page 10 in the main manuscript, the authors stated “..., nor find any decay of the entire signal as a function of the field strength due to the tunneling to higher energy bands at intense field regime.” I agree to the authors that the real excitation of carriers cannot explain the data due to the absence of the signal decay. However, the virtual population generated only under the THz-field irradiation should induce additional transitions for probe pulses, which would result in induced absorption signals. Is it possible to comment on this?

→The original manuscript contained a paragraph on the possible carrier generation by THz (multi-photon processes and impact ionization) in our method section and also the Supplementary figure:

“In this linear-optical regime, we thus neglect carrier generation by multi-photon processes and impact ionization, which does not seem to play a dominant role in the measured transient spectra. Interband tunneling by the THz field could lead to bleaching at later delay times and the slightly

asymmetric spectral evolution with respect to $\tau = 0$ (Fig. 2(A)) (corresponding to the trailing edge of the THz transient in the Supplementary Material of ref [20]). However, significant carrier multiplication does not occur within this experimental window, as shown in Supplementary Fig. 10.”

However, in addition to that, we have performed new simulations that include the possible effects of THz -induced generation of virtual and real carriers that may arise from multi-photon transitions by solving a full set of semiconductor Bloch equations, in which the THz field is treated non-perturbatively, and the weak optical probe pulse is considered linearly. The result is shown in the supplementary fig 6, indicating that for the perovskite, the high-order/multi-photon interband effects are basically negligible. Therefore, our simplified model defined by Eq. (1) in the main text captures all the relevant physics. Accordingly, we have added the following statement in the main text:

“Furthermore, to analyze the possible effects of THz-induced generation of virtual and real carriers that may arise from multi-photon transitions, we solve a full set of SBE (Supplementary Methods) in which the THz field is included non-perturbatively and the weak optical probe pulse is considered linearly. The results (Supplementary Figure 6) show that for the considered field amplitudes, such higher-order interband effects arising from the THz fields are negligible, as the results from the full equations are very close to the ones obtained from the simplified Eq. (1).”

REVIEWERS' COMMENTS

Reviewer #3 (Remarks to the Author):

I appreciate the efforts of the authors to answer my questions and comments. The authors have cleared up my concerns about the interpretation of the observed nonlinear responses by performing new calculations (Supplementary Figure 6) and ruling out the possible interpretations other than the Wannier-Stark localization. Therefore, now I think the manuscript is suitable to publish in Nature Communications.

Reviewer #3 (Remarks to the Author):

I appreciate the efforts of the authors to answer my questions and comments. The authors have cleared up my concerns about the interpretation of the observed nonlinear responses by performing new calculations (Supplementary Figure 6) and ruling out the possible interpretations other than the Wannier-Stark localization. Therefore, now I think the manuscript is suitable to publish in Nature Communications.

→ We thank the reviewer very much for the thorough reviews and constructive comments.